# BMAL1 coordinates energy metabolism and differentiation of pluripotent stem cells

Cristina Ameneiro[1], Tiago Moreira[1], Alejandro Fuentes-Iglesias[1,2],
Alba Coego[1], Vera Garcia-Outeiral[1,2], Adriana Escudero[1,2],
Daniel Torrecilla[1], Sonia Mulero-Navarro[3], Jose Maria Carvajal-Gonzalez[3],
Diana Guallar[1,4], Miguel Fidalgo[1,2]

**BMAL1 is essential for the regulation of circadian rhythms in differentiated cells and adult stem cells, but the molecular underpinnings of its function in pluripotent cells, which hold a great potential in regenerative medicine, remain to be addressed. Here, using transient and permanent loss-of-function approaches in mouse embryonic stem cells (ESCs), we reveal that although BMAL1 is dispensable for the maintenance of the pluripotent state, its depletion leads to deregulation of transcriptional programs linked to cell differentiation commitment. We further confirm that depletion of *Bmal1* alters the differentiation potential of ESCs in vitro. Mechanistically, we demonstrate that BMAL1 participates in the regulation of energy metabolism maintaining a low mitochondrial function which is associated with pluripotency. Loss-of-function of *Bmal1* leads to the deregulation of metabolic gene expression associated with a shift from glycolytic to oxidative metabolism. Our results highlight the important role that BMAL1 plays at the exit of pluripotency in vitro and provide evidence implicating a non-canonical circadian function of BMAL1 in the metabolic control for cell fate determination.**

## Introduction

Circadian rhythms are necessary to coordinate key behavioural (e.g., sleep/wake cycle) and physiological (e.g., metabolism, hormone secretion, and stem cell homeostasis) processes in mammals (Bechtold & Loudon, 2013; Lopez-Minguez et al, 2016; McAlpine & Swirski, 2016; Weger et al, 2017; Dierickx et al, 2018). At the cellular level, the circadian clock is composed by transcriptional and translational feedback loops involving the clock master regulators BMAL1, CLOCK, PER, and CRY proteins, which ensure rhythmic gene expression to accommodate to the tissue and organ needs. Interestingly, although the proteins of the circadian clock are already present at early stages of embryonic development, circadian rhythms are not established until around the mid-gestation stage (Saxena et al, 2007; Umemura et al, 2017). In line with this, embryonic stem cells (ESCs), which are derived from the inner cell mass of the preimplantation blastocyst, are devoid of transcriptional circadian oscillations (Kowalska et al, 2010; Yagita et al, 2010; Umemura et al, 2014, 2017; Dierickx et al, 2017).

Given the lack of a compensating homologue in vivo, BMAL1 has been defined as the only essential component of the molecular circadian clock in mammals (Bunger et al, 2000). *Bmal1* KO mice have impaired circadian behaviour and absence of rhythmicity in circadian target genes (Bunger et al, 2000). Moreover, they show infertility (Alvarez et al, 2008; Boden et al, 2010), show impaired glucose homeostasis (Rudic et al, 2004), and have been reported to have reduced life span and higher prevalence of age-related pathologies (Kondratov et al, 2006). Unexpectedly, many metabolic and age-related pathologies caused by *Bmal1* depletion were not observed when using an inducible KO mouse model where *Bmal1* depletion was performed in the adult age (Yang et al, 2016), suggesting important functions for this master regulator during embryogenesis. Given that BMAL1 is readily expressed in ESCs, even in the absence of a functional circadian clock, we hypothesized that additional roles of this factor in pluripotency remain to be discovered and could yield insights into its function during early stages of embryonic development.

To investigate the function of BMAL1 in pluripotent cells, which present a great therapeutic potential given their ability to generate cells of any adult tissue, we used transient and genetic models of *Bmal1* loss-of-function in ESCs. We discovered that BMAL1 is dispensable for ESC maintenance, as its depletion does not affect pluripotency marker expression or colony formation. Nevertheless, we observed that ablation of *Bmal1* in ESCs resulted in deregulation of genes from the three embryonic germ layers, and an aberrant

[1]Center for Research in Molecular Medicine and Chronic Diseases (CIMUS), Universidade de Santiago de Compostela (USC)–Health Research Institute (IDIS), Santiago de Compostela, Spain   [2]Department of Physiology, USC, Santiago de Compostela, Spain   [3]Department of Biochemistry, Molecular Biology and Genetics, Facultad de Ciencias, Universidad de Extremadura, Badajoz, Spain   [4]Department of Biochemistry and Molecular Biology, USC, Santiago de Compostela, Spain

Correspondence: miguel.fidalgo@usc.es; diana.guallar@usc.es

 

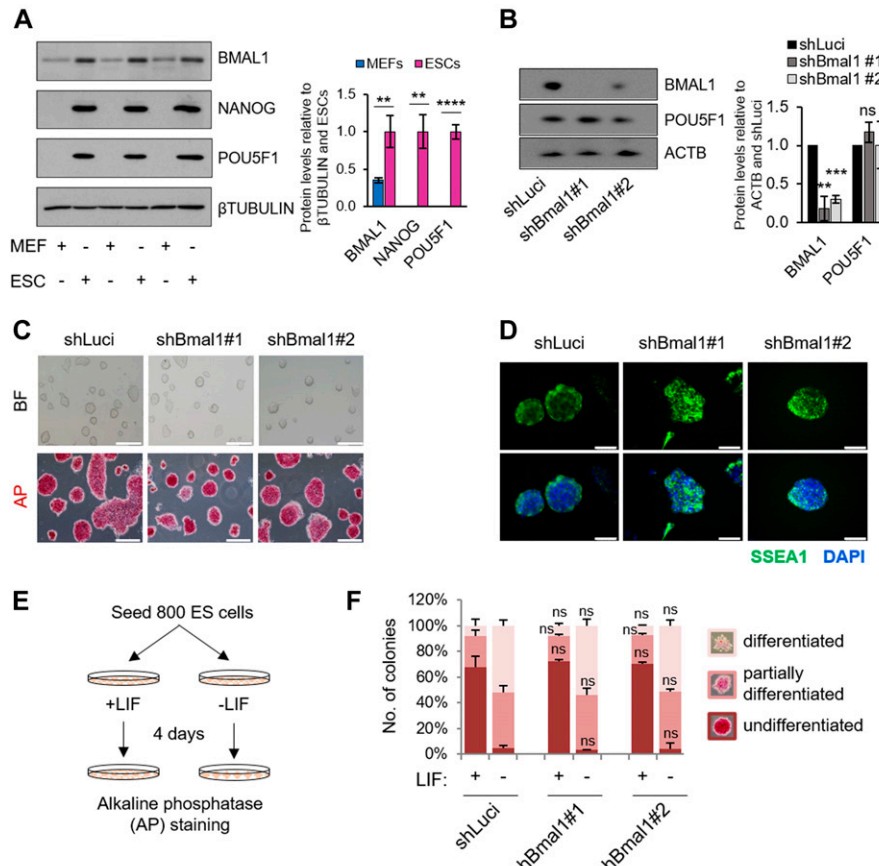

**Figure 1. Silencing of *Bmal1* does not compromise the maintenance of embryonic stem cells (ESCs).**
**(A)** (Left) Western blot of BMAL1 and the pluripotency markers NANOG and POU5F1 in MEFs and ESCs. βTUBULIN was used as loading control. Biological triplicates are shown. (Right) Quantification of the blots. **(B)** (Left) Western blot of BMAL1 and POU5F1 in ESCs transfected with shRNAs against Bmal1 or Luciferase as control. ACTIN (ACTB) was used as a loading control. One representative experiment is shown. (Right) Quantification of three independent experiments. **(C)** Bright-field (BF) and AP-staining images of ESCs transfected with shRNA against Bmal1 or Luciferase as a control. The scale bar represents 200 μm. **(D)** Immunofluorescence of the pluripotent marker SSEA1 in ESCs transfected with shRNAs against Bmal1 or Luciferase as control. Nuclei were stained with DAPI. The scale bar represents 50 μm. **(E)** Schematic representation for the colony formation assay. ESCs transfected with shRNA against Bmal1 or Luciferase as control are seeded at low confluence in standard medium with or without LIF, and colonies are counted and classified into the three indicated categories according to their AP-staining intensity. **(F)** Graphic representation for the counting and classification of colonies. (n = 3). In (A, B, F), bars represent mean ± SD. Two-tailed unpaired *t* test analysis was performed. ****P < 0.0001, ***P < 0.001, **P < 0.01; ns, not significant. Source data are available for this figure.

induction of differentiation gene expression in vitro. Importantly, using embryonic organoids, we discovered that BMAL1 is necessary for in vitro gastruloid formation and proper expression of lineage specification markers. Mechanistically, we discovered that depletion of *Bmal1* produced a change in metabolism-related genes and pathways, which are now considered to be drivers in the differentiation process. In particular, we observed a reduction in basal glycolysis and a concomitant increase in respiration, which was accompanied by an increase in mitochondrial reactive oxygen species (mtROS) production. Thus, our results uncover an unexpected function of BMAL1 in ESCs in metabolic regulation, where the clock is not yet "ticking," but BMAL1 function is already relevant for proper embryonic specification.

## Results

### Transient loss-of-function of BMAL1 is dispensable for ESC self-renewal

To define the role of BMAL1 in pluripotent cells, which have been previously reported to lack circadian rhythms (Kowalska et al, 2010; Yagita et al, 2010; Umemura et al, 2014, 2017; Dierickx et al, 2017), we

first determined the expression level of this core clock regulator in MEFs and pluripotent ESCs. Notably, when we analysed BMAL1 RNA and protein levels by RT quantitative PCR (RT-qPCR) and Western blot, respectively, we detected that its abundance was higher in ESCs compared with MEFs (Figs 1A and S1A). In contrast, expression of *Clock*, another circadian regulator, was higher in MEFs, which have been reported to possess a functional circadian clock (Yagita et al, 2001) (Fig S1A). These data prompted us to consider whether BMAL1 may be important for pluripotency.

To understand the role of BMAL1 in ESCs, we first performed loss-of-function experiments using two independent shRNAs against Bmal1 to reduce the likelihood of off-target effects. We confirmed the efficiency in silencing of Bmal1 compared with Luciferase control (shLuci) knockdown by analysing its protein levels by Western blot (Fig 1B). Importantly, we did not observe major differences in typical ESC morphology and AP staining in cells transduced with shRNAs against Bmal1 compared with shLuci (Fig 1C). Consistently, reduction of BMAL1 protein levels did not greatly impact pluripotency markers (i.e., *Pou5f1* and *Nanog*) at the mRNA level (Fig S1B) and/or the protein level (Fig 1B). Indeed, immunofluorescence assays revealed that silencing of Bmal1 does not alter expression variability and population heterogeneity of pluripotency markers (i.e., SSEA1, POU5F1, and ZFP281) in ESCs (Figs 1D and S1C). We next examined whether self-renewal properties of pluripotent cells were affected by the loss-of-function of *Bmal1* in the presence

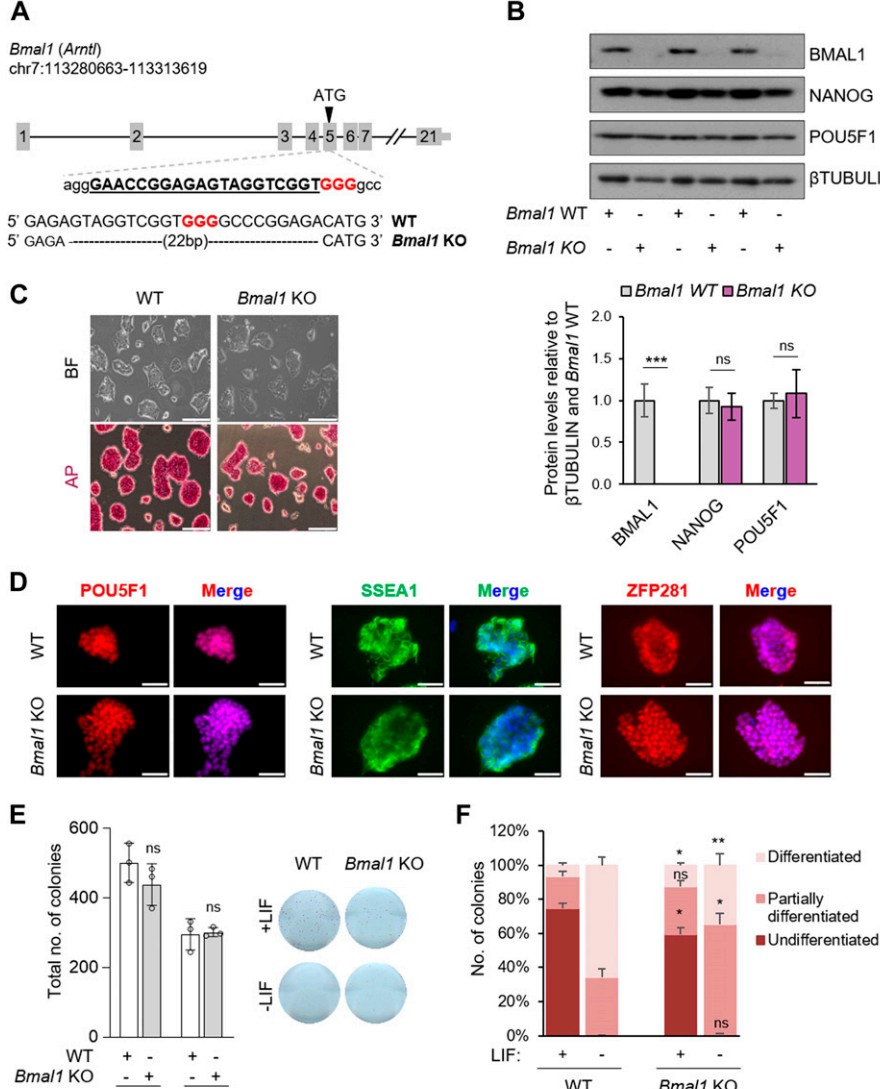

**Figure 2. Generation and characterization of a CRISPR/Cas9 *Bmal1* KO embryonic stem cell (ESC) line.**
**(A)** (Top) Schematic representation for the CRISPR/Cas9 strategy used. The designed sgRNA is underlined and the PAM sequence is highlighted in red. (Bottom) Sequence deleted (22 bp) in *Bmal1* KO alleles detected by Sanger sequencing. **(B)** (Top) Western blot of BMAL1 and the pluripotency factors NANOG and POU5F1 in *Bmal1* WT and KO ESCs. βTUBULIN was used as a loading control. (Bottom) Quantification of three independent experiments. **(C)** Bright-field (BF) and AP-staining images of *Bmal1* WT and KO ESCs. The scale bar represents 200 μm. **(D)** Immunofluorescence of POU5F1, SSEA1, and ZFP281 in *Bmal1* WT and KO ESCs. Nuclei were stained with DAPI. The scale bar represents 50 μm. **(E)** (Left) Number of ESC colonies formed in a clonogenic assay with *Bmal1* WT and KO ESCs. (Right) Representative images of AP-stained wells are shown. **(F)** Graphic representation for the counting and classification of colonies of a colony-formation assay. (n = 3) In (B, E, F), bars represent mean ± SD. Two-tailed unpaired *t* test analysis was performed. ***P < 0.001, **P < 0.01, *P < 0.05; ns, not significant. Source data are available for this figure.

or absence of leukaemia inhibitory factor (LIF). For this purpose, a reduced number of cells transduced with either the control shLuci or each of the two shRNAs against Bmal1 were plated and cultured at clonal density with or without LIF for 4 d, followed by AP staining (Fig 1E). Consistent with the effect observed in bulk-grown ESCs, silencing of Bmal1 did not impact the proportion of undifferentiated, partially or fully differentiated colonies both in pluripotency-sustaining (+LIF) or differentiation-promoting (−LIF) culture conditions (Fig 1F). Taken together, our results show that BMAL1 is dispensable for ESC self-renewal, despite being abundantly expressed in pluripotent cells.

## Absence of BMAL1 protein is compatible with pluripotency maintenance

Given that knockdown with RNA interference can result in variable amounts of mRNA reduction, we aimed at generating ESC lines completely lacking *Bmal1* expression to systematically dissect BMAL1 function in pluripotent cells. To this end, we used the CRISPR-Cas9 nuclease system (Jinek et al, 2012; Cong et al, 2013) to generate a *Bmal1* KO ESC line. First, we designed a sgRNA specifically targeting the start codon site of *Bmal1* gene, located at the Exon 5 (Fig 2A). PCR genotyping and Sanger sequencing identified a *Bmal1* KO ESC line (Figs 2A and S2A–C). Western blot analysis confirmed complete depletion of BMAL1 protein in our selected ESC clone (Fig 2B).

To establish whether depletion of *Bmal1* influences the maintenance of the pluripotent state of ESCs, we first examined the morphology and AP-staining pattern in *Bmal1* KO ESCs. Notably, consistent with the Bmal1 knockdown data, *Bmal1* KO ESCs displayed normal pluripotent colony morphology and stained positive for AP compared with WT pluripotent cells over the course of multiple passages (>10) (Fig 2C). In addition, the expression of pluripotency markers (i.e., *Nanog*, *Pou5f1*, and *Zfp42*) at the RNA and protein levels was not significantly affected in the absence of BMAL1

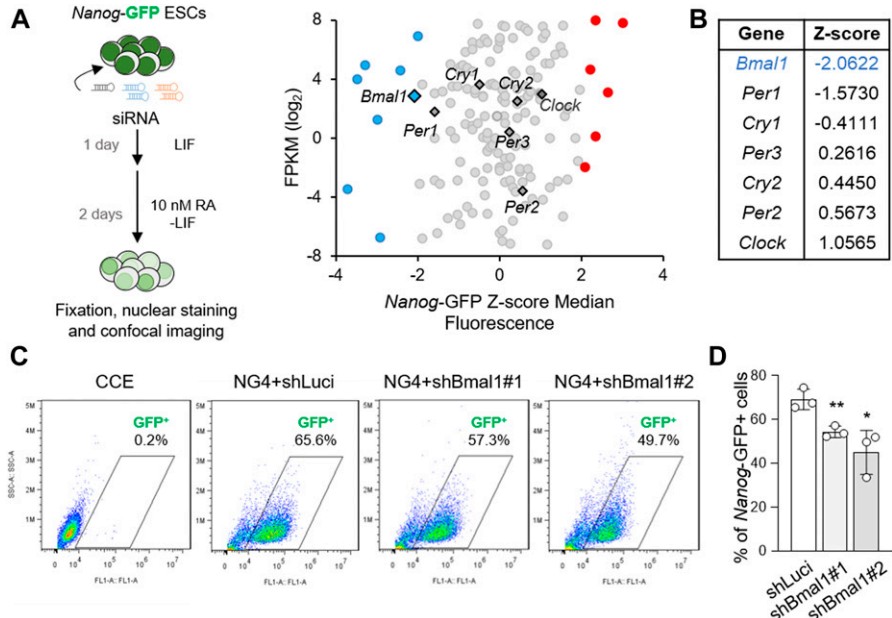

**Figure 3. Absence of Bmal1 affects the exit from pluripotency.**
**(A)** (Left) Schematic representation for screening conditions. *Nanog*-GFP (NG4) ESCs are transfected with siRNAs and cultured for 1 d in standard pluripotency medium followed by 2 d under mild retinoic acid (RA)–mediated differentiation. The cells are fixed, stained, and imaged. (Right) Scatter plot showing circadian clock core genes expression represented as FPKM and their effect in *Nanog*-GFP median fluorescence after screening conditions. **(B)** Summary table of Z-score median fluorescence values for circadian clock core genes after screening conditions highlighting *Bmal1* as the gene with the highest effect among the core circadian regulators. **(C)** FACS analysis of *Nanog*-GFP intensity after RA-mediated differentiation of NG4 cells transfected with shRNA against Bmal1 or Luciferase as control. CCE parental ESC line is used as negative fluorescence control. The percentage represents the fraction of *Nanog*-GFP–positive cells in each sample in the indicated gate. One representative experiment is shown. **(D)** Percentage of *Nanog*-GFP–positive cells after transduction with the indicated shRNAs. (n = 3) Bars represent mean ± SD. Two-tailed unpaired *t* test analysis was performed. **P < 0.01, *P < 0.05.
Source data are available for this figure.

(Figs 2B and S2D and E). Moreover, immunofluorescence of key pluripotent markers (i.e., POU5F1, SSEA1, ZFP281, SOX2, and NANOG) (Figs 2D and S2F) suggested that pluripotency maintenance in *Bmal1* KO ESCs was grossly unaffected. On the other hand, when we analysed the self-renewal capability of *Bmal1*-depleted ESCs, we did not observe changes in the number of colonies (Fig 2E) and only a mild reduction in the percentage of undifferentiated colonies (74.5 ± 3.3 versus 59.6 ± 3.9) in the presence of LIF (Fig 2F), which is in line with the acute silencing of Bmal1 using the shRNA approach (Fig 1F). Of note, *Bmal1* KO ESCs showed a defect in differentiation upon LIF withdrawal as observed by the decrease in the percentage of fully differentiated colonies (65.6 ± 4.7 versus 34.7 ± 6.7) (Fig 2F), suggesting a possible role of BMAL1 during the exit of pluripotency. Collectively, taking together the knockdown and KO results, our data demonstrate that BMAL1 is dispensable for ESC maintenance.

### BMAL1 is required for exiting pluripotency in vitro

Given that pluripotent cells can differentiate into all cell types from the three germ layers (i.e., ectoderm, endoderm, and mesoderm), we next interrogated the role of BMAL1 during the cellular differentiation process of ESCs. For this purpose, we first performed an in silico analysis using a genome-wide RNAi screen–published dataset for the identification of potential direct and indirect regulators of *Nanog* gene expression under mild retinoic acid (RA)–induced differentiation conditions, which could presumably be involved in the exit of pluripotency (Gingold et al, 2014). Remarkably, when we analysed the consequences of loss-of-function in 166 genes involved with circadian rhythms (GO:0007623), we noticed the existence of eight positive, including BMAL1, and six negative potential regulators of the expression of the pluripotency marker *Nanog* (Z score > 2), measured by GFP fluorescent levels, which could play roles in the differentiation process (Fig 3A). Interestingly, we also observed the existence of

opposing effects on *Nanog* promoter activity between the core circadian clock regulators BMAL1 and CLOCK (Fig 3B). Importantly, we further confirmed the regulatory effect of BMAL1 on *Nanog* promoter activity under the same RA-induced differentiation conditions, by flow cytometry analysis using the *Nanog*-GFP reporter ESC line NG4 (Schaniel et al, 2009) transduced with shRNAs against Bmal1 and shLuci as a control (Fig 3C and D). Future studies are warranted to investigate whether BMAL1 regulates *Nanog* promoter activity in a direct or indirect manner. Taken together, these results show that BMAL1 may be involved in early decisions during the exit of pluripotency and unveil the existence of specific functions of factors related with circadian processes in cell fate determination at the beginning of in vitro differentiation.

Next, to investigate the function of BMAL1 in differentiation, we analysed the ability of *Bmal1* KO and wild-type (WT) ESCs to form teratomas. When subcutaneously injected in the flanks of immunodeficiency mice, both *Bmal1* KO and WT ESCs were able to efficiently form teratomas of similar size (Fig 4A). Haematoxylin and eosin–stained sections of teratomas showed that a range of cell types and tissues from all three germ layers (i.e., mesoderm, ectoderm, and endoderm) was present in all teratomas regardless of BMAL1 presence (Fig 4B). These findings, together with our results showing that BMAL1 is dispensable for ESC maintenance, support that *Bmal1* KO ESCs are pluripotent. It is possible that the role of BMAL1 in the exit of pluripotency observed in vitro (Fig 3) does not cause profound abnormalities which would be detectable during differentiation in vivo by teratoma assay. Thus, to determine if depletion of *Bmal1* in ESCs can affect cellular differentiation in vitro, we performed embryoid bodies (EBs) assays, which mimic early embryonic development *in a dish* (Doetschman et al, 1985). After 6 d of culture in ESC medium without the LIF cytokine (Fig 4C), we observed that *Bmal1* expression was down-regulated similarly to *Nanog*, concomitant with the up-regulation of markers from the three germ layers (i.e., *Pax3*, *Gata6*, and *Mixl1*) (Fig 4D). Consistent

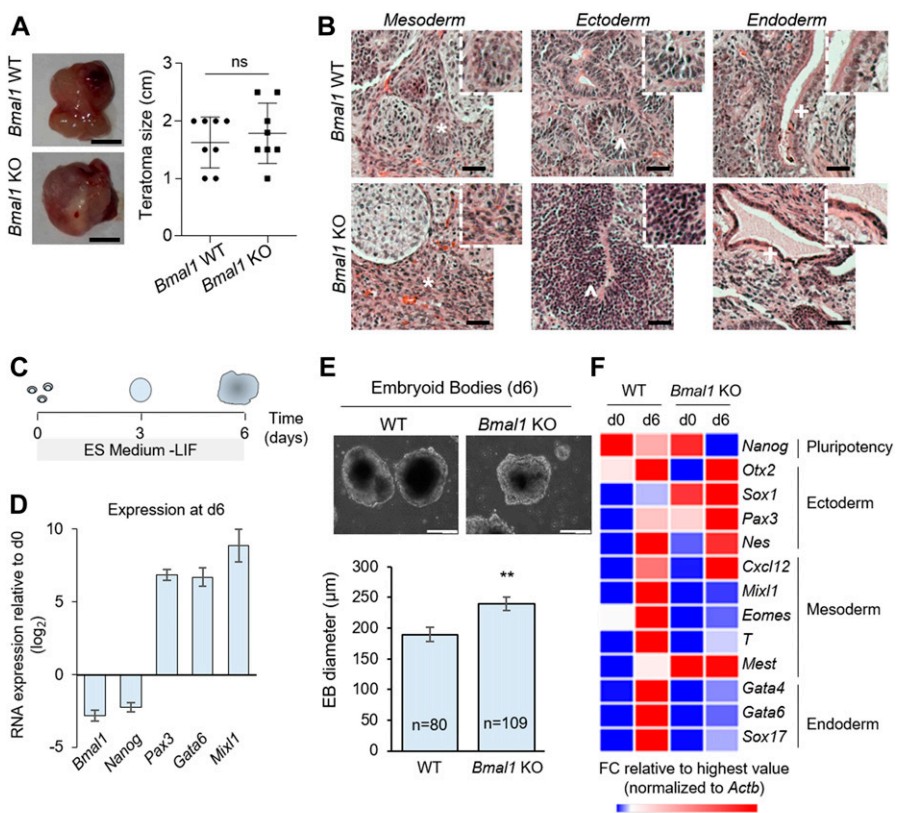

**Figure 4. *Bmal1* is required for embryonic stem cell in vitro differentiation.**
**(A)** (Left) Teratomas formed by wild-type (WT) and *Bmal1* KO ES cells after 3–4 wk of injection. The bar represents 5 mm. (Right) Quantification of teratoma size (n = 8 tumours per genotype). **(B)** Histological analysis of teratomas of the indicated genotype by H&E staining. Structures representing the three embryonic germ layers in both genotypes were found. (\*), connective tissue; (^), neural tissue; and (+), epithelial tissue. 4× magnifications for each tissue are shown. The bar represents 30 µm. **(C)** Schematic representation for the embryoid body (EB) differentiation assay. **(D)** Relative expression of genes at day 6 of EB differentiation assay compared with day 0 in WT embryonic stem cells determined by RT-qPCR. Ectoderm-specific (i.e., *Pax3*), endoderm-specific (i.e., *Gata6*), and mesoderm-specific (i.e., *Mixl1*) markers are shown. **(E)** (Top) Representative images of EBs at day 6 of differentiation in the indicated cell lines. The scale bar represents 200 µm. (Bottom) Quantification of the diameter of day 6 EBs for *Bmal1* WT and KO cells. The number of EBs analysed is indicated. Data are shown as mean ± SEM. **(F)** Relative expression of several germ layer–specific genes from day 0 and day 6 EBs determined by RT-qPCR. Expression fold changes are shown relative to the highest value. In (A, D), data are shown as mean ± SD. Two-tailed unpaired *t* test was performed. \*\**P* < 0.01; ns, not significant.
Source data are available for this figure.

with their ability to generate teratomas, *Bmal1* KO cells were able to form EBs (Fig 4E). However, we observed that *Bmal1* KO EBs presented a significant increase in their size compared with the control (Fig 4E), suggesting intrinsic differences during the differentiation process through EB formation due to the absence of this master clock regulator. Moreover, when we analysed gene expression of several ectoderm, mesoderm, and endoderm markers, we found that they were differentially induced in *Bmal1* KO at day 6, compared with WT EBs (Fig 4F). These results indicate that BMAL1 is required for ESC differentiation in vitro to properly establish germ layer–specific transcriptional programs.

## BMAL1 is important during in vitro gastrulation

To further confirm our observation that loss of BMAL1 affects proper embryonic germ layer specification in vitro, we decided to use a recently reported gastruloid system (Beccari et al, 2018). Gastruloids are small aggregates of ESCs that undergo gastrulation-like events and elongation in vitro and mimic embryonic spatial and temporal gene expression (Beccari et al, 2018). Importantly, this gastrulation model can be used as an in vitro system to study early developmental events taking place in the mammalian embryo in vivo. Thus, we generated aggregates of WT or *Bmal1* KO ESCs in N2B27 medium and subjected them to a pulse of a WNT agonist (i.e., CHIR99021) (Fig 5A). First, we analysed the expression pattern of WT gastruloids and observed that, similar to the pluripotency marker *Nanog*, *Bmal1* was down-regulated at 120 h of gastruloid formation compared with ESCs (t = 0 h),

concomitant with germ layer marker induction (i.e., *Pax3*, *Gata6*, and *Mixl1*) (Fig 5B). We then compared the efficiency in the generation of gastruloids in the presence or absence of BMAL1. Remarkably, we observed that after 120 h, the aggregates obtained from *Bmal1* KO ESCs were significantly smaller (Fig 5C and D) and failed to elongate and polarize to give rise to gastruloid-like structures compared with wild-type cells (7.14% versus 41.07%, respectively) (Fig 5E). In addition, absence of BMAL1 during gastruloid formation was accompanied by altered expression of several lineage specification markers representative of three germ layers (Fig 5F). In particular, we observed deregulation, in the absence of BMAL1, of genes associated with in vivo gastrulation process such as *Mixl1* and *Eomes* (Beccari et al, 2018). Likewise, we observed altered transcriptional dynamics of several members of the *Hoxd* gene cluster (Fig 5G), which is one of the hallmarks of axial gene regulatory systems whose sequential activation is associated with the patterning and formation during in vitro gastruloid formation (Beccari et al, 2018). Collectively, these results show that BMAL1 is required for efficient gastruloid formation in vitro and confirm that BMAL1 deficiency abrogates the correct induction of ectoderm, mesoderm, and endoderm markers during the exit of pluripotency.

## BMAL1 regulates transcriptional networks related to cellular differentiation

To gain further insight into the molecular underpinnings of BMAL1 function in pluripotent cells, we performed transcriptional profiling

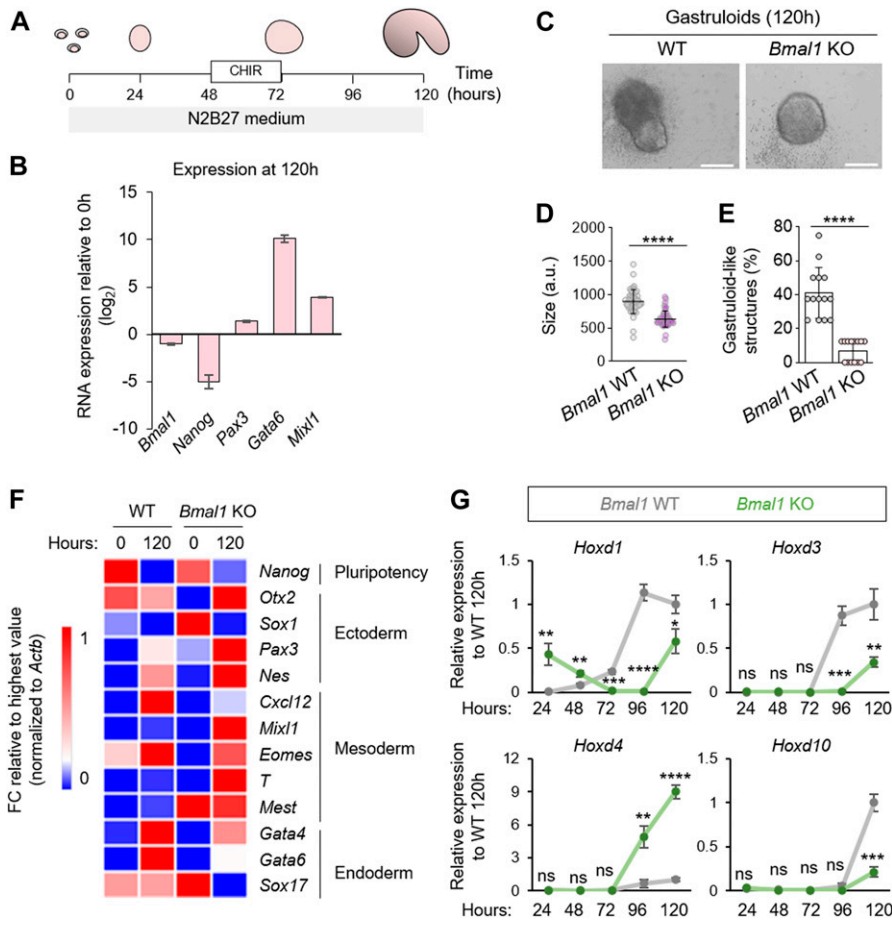

**Figure 5. Ablation of *Bmal1* disrupts gastrulation in vitro.**
**(A)** Schematic depiction for the gastrulation assay performed (Beccari et al, 2018) for CHIR, GSK-3 inhibitor. **(B)** Relative expression of genes after 120 h of in vitro gastrulation assay of wild-type embryonic stem cells (ESCs) determined by RT-qPCR and expressed relative to ESCs (0 h). Ectoderm-specific (i.e., *Pax3*), endoderm-specific (i.e., *Gata6*), and mesoderm-specific (i.e., *Mixl1*) markers are shown. **(C)** Representative images of 120 h aggregates for *Bmal1* WT and KO cells. The scale bar represents 200 *µm*. **(D)** Quantification of the size of the aggregates formed by *Bmal1* WT and KO cells at 120 h of in vitro gastrulation. **(E)** Percentage of gastruloid-like structures formed by *Bmal1* WT and KO aggregates at 120 h of in vitro gastrulation (n = 14 independent experiments with at least eight formed aggregates per experiment). **(F)** Relative expression of *Nanog* and several germ layer–specific genes at the indicated time points of the gastruloid differentiation assay of WT and *Bmal1* KO ESCs determined by RT-qPCR. Expression of fold changes are shown relative to the highest value. **(G)** Expression of *Hoxd1*, *Hoxd3*, *Hoxd4*, and *Hoxd10* dynamics during in vitro gastrulation of *Bmal1* WT and KO ESCs at the indicated time points. (n = 3). In (B, D, E, G), data are represented as mean ± SD. Two-tailed unpaired *t* test was performed. ****P < 0.0001, ***P < 0.001, **P < 0.01; a.u., arbitrary units; ns, not significant.
Source data are available for this figure.

of wild-type and *Bmal1* KO ESCs. We identified 444 up-regulated and 197 down-regulated genes that were significantly changed with a difference in abundance greater than twofold in the absence of BMAL1 (Fig 6A and Table S1). Remarkably, Gene Ontology (GO) analysis of these misregulated genes upon *Bmal1* depletion showed enrichment for differentiation processes (Fig 6B). To validate these results, we performed RT-qPCR analysis of a number of up-regulated (e.g., *Tead4* and *Mest*) and down-regulated (e.g., *Eomes* and *Snai3*) early cell fate markers upon *Bmal1* KO in ESCs (Fig 6C and D). Moreover, gene set enrichment analysis (GSEA) also identified the up-regulation of gene signatures related to stem cell differentiation in the absence of BMAL1 function when compared with their wild-type counterparts (Fig 6E). Thus, these results suggest that *Bmal1* depletion leads to transcriptional changes in genes related to developmental processes by direct or indirect mechanisms, which is in line with our results showing the requirement of this circadian master regulator for proper cellular differentiation of ESCs. Indeed, we observed that depletion of *Bmal1* led to a significant deregulation of genes related to the three embryo germ layers (i.e., endoderm, ectoderm, and mesoderm) in ESCs, whereas housekeeping genes remained unaltered (Fig 6F). Importantly, in spite of these global transcriptional changes, the GSEA analysis further confirmed the pluripotent cell identity of *Bmal1* KO ESCs denoted by no significant changes in the expression

of ESC-enriched genes as well as targets of the core pluripotency regulators POU5F1 (OCT4), SOX2, and NANOG (OSN) (Fig S3A). Taken together, our data suggest that BMAL1 loss may influence the differentiation potential of ESCs in vitro by altering the expression of early specification genes of the three germ layers in pluripotent cells.

### BMAL1 supports glycolytic metabolism in ESCs

To understand how BMAL1 orchestrates transcriptional programs involved in proper cell differentiation in vitro, we next compared the gene expression changes after *Bmal1* depletion in ESCs with those caused by loss-of-function of other transcription regulators using the Network2Canvas computational tool (Tan et al, 2013). Interestingly, we found a close correlation of the transcriptional changes in the absence of BMAL1 with those upon depletion of the master pluripotency regulator POU5F1 (Kim et al, 2015) as well as other pluripotent regulators related with differentiation and/or cellular metabolism, including ZFX (Galan-Caridad et al, 2007; Chen et al, 2008) and Polycomb members (i.e., EED and SUZ12) (Brookes et al, 2012; Di Croce & Helin, 2013) (Fig S3B). These results, together with the observation that the expression of these factors was not altered upon *Bmal1* depletion in mouse ESCs (Figs 2B and S3C and D), suggest the existence of common pathways regulated

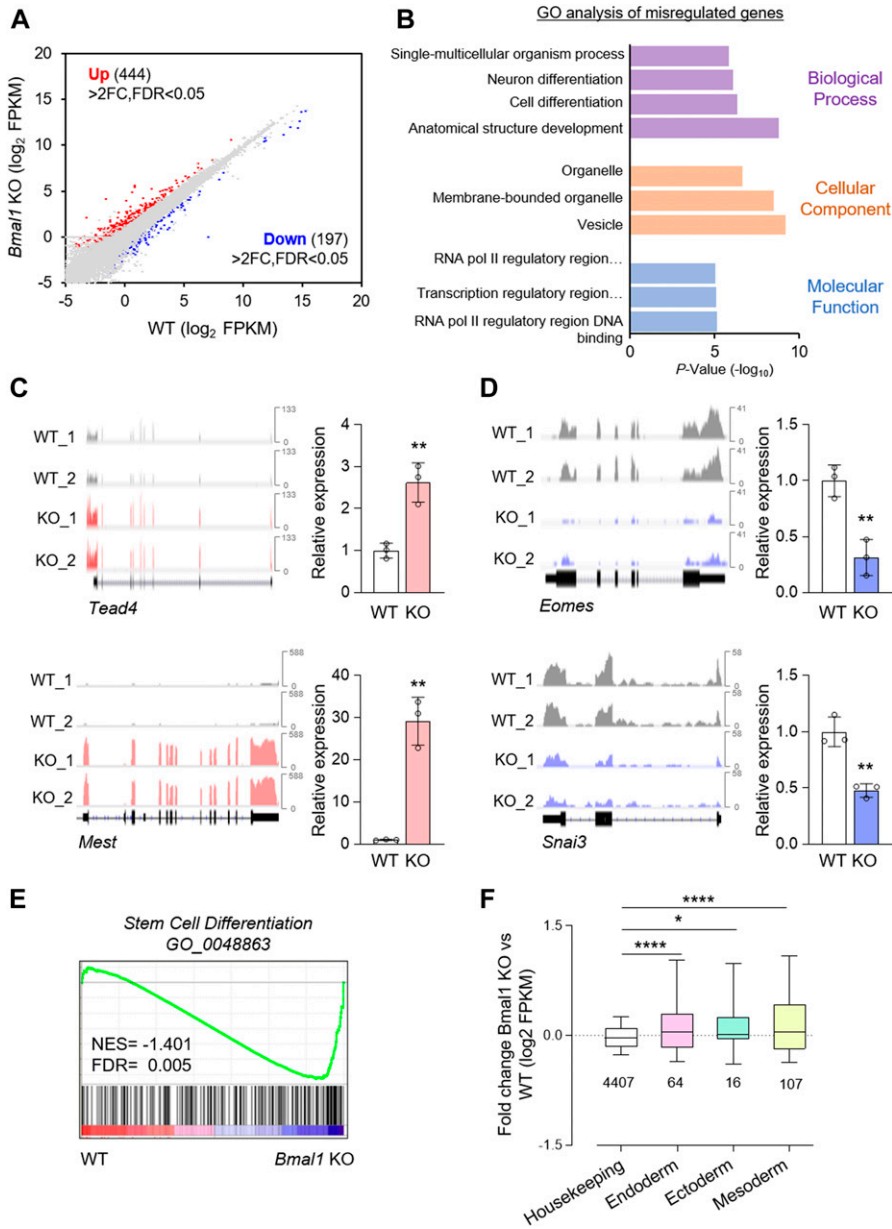

**Figure 6. *Bmal1* participates in the regulation of cell differentiation transcriptional programs in embryonic stem cells (ESCs).**

**(A)** Scatter plot of gene expression (in $\log_2$ FPKM) in *Bmal1* KO versus wild-type (WT) control ESCs determined by RNA-seq. Biological duplicates were analysed. Genes significantly (false discovery rate [FDR] < 0.05) up-regulated and down-regulated more than twofold are shown in red and blue, respectively. **(B)** Gene Ontology (GO) analysis of genes misregulated in *Bmal1* KO ESCs. The x-axis corresponds to the negative $\log_{10}$ P-values. **(C)** (Left panel) Integrative genome viewer representation of RNA-seq reads for representative up-regulated genes in *Bmal1* KO cells. (Right panel) RT-qPCR analysis validating the RNA-seq results (n = 3). **(D)** (Left panel) Integrative genome viewer representation of RNA-seq reads for representative down-regulated genes in *Bmal1* KO cells. (Right panel) RT-qPCR analysis validating the RNA-seq results (n = 3). **(E)** Gene set enrichment analysis from WT and *Bmal1* KO RNA-seq data against a stem cell differentiation gene set. Normalized enrichment score (NES) and FDR are shown. **(F)** Fold RNA expression changes of the indicated gene categories in WT and *Bmal1* KO ESCs. The number of genes included in each category is indicated. Whiskers extend to the $10^{th}$–$90^{th}$ percentile range. In (C, D), data are shown as mean ± SD. Two-tailed unpaired *t* test was performed. ****$P$ < 0.0001, **$P$ < 0.01, *$P$ < 0.05. Source data are available for this figure.

by this set of factors and BMAL1. Indeed, Kyoto Encyclopedia of Genes and Genomes (KEGG) pathway analysis of the misregulated genes in *Bmal1* KO ESCs revealed an enrichment in metabolic-associated processes (Fig S4A). Notably, among the most differentially expressed genes (false discovery rate [FDR] > 0.05 and fold change > 2), we found 19 down-regulated and 49 up-regulated metabolism-related genes in *Bmal1*-depleted cells (Fig S4B). In particular, we observed that BMAL1 loss influences the expression of several genes related with mitochondrial complex I–V and the tricarboxylic acid cycle or Krebs cycle (Figs 7A and S4C). Collectively, our results show that BMAL1 contributes to the proper transcriptional landscape regulation of pluripotent cells and that its depletion leads to the deregulation of metabolism-related transcriptional pathways. Changes in metabolic activity are closely linked to the exit of

pluripotency, partly by influencing the epigenome during cell commitment (Cliff & Dalton, 2017; Dahan et al, 2019). Thus, we speculated that *Bmal1* depletion can alter early cell differentiation potential through changes in the expression of metabolic gene networks that govern the balance between glycolytic and oxidative phosphorylation (OXPHOS) activity. In agreement with our hypothesis, we found that *Bmal1* KO cells showed reduced basal glycolysis compared with WT cells (Fig 7B). Conversely, depletion of *Bmal1* in ESCs led to an increase in the basal oxygen consumption rate (OCR) (Fig 7C). Thus, we observed a metabolic switch in ESCs caused by *Bmal1* depletion, which was translated into a more oxidative versus glycolytic use of glucose in *Bmal1* KO ESCs compared with WT ones (Fig 7D). Although there is usually a tight coupling between electron transport and ATP synthesis, under certain conditions, protons can

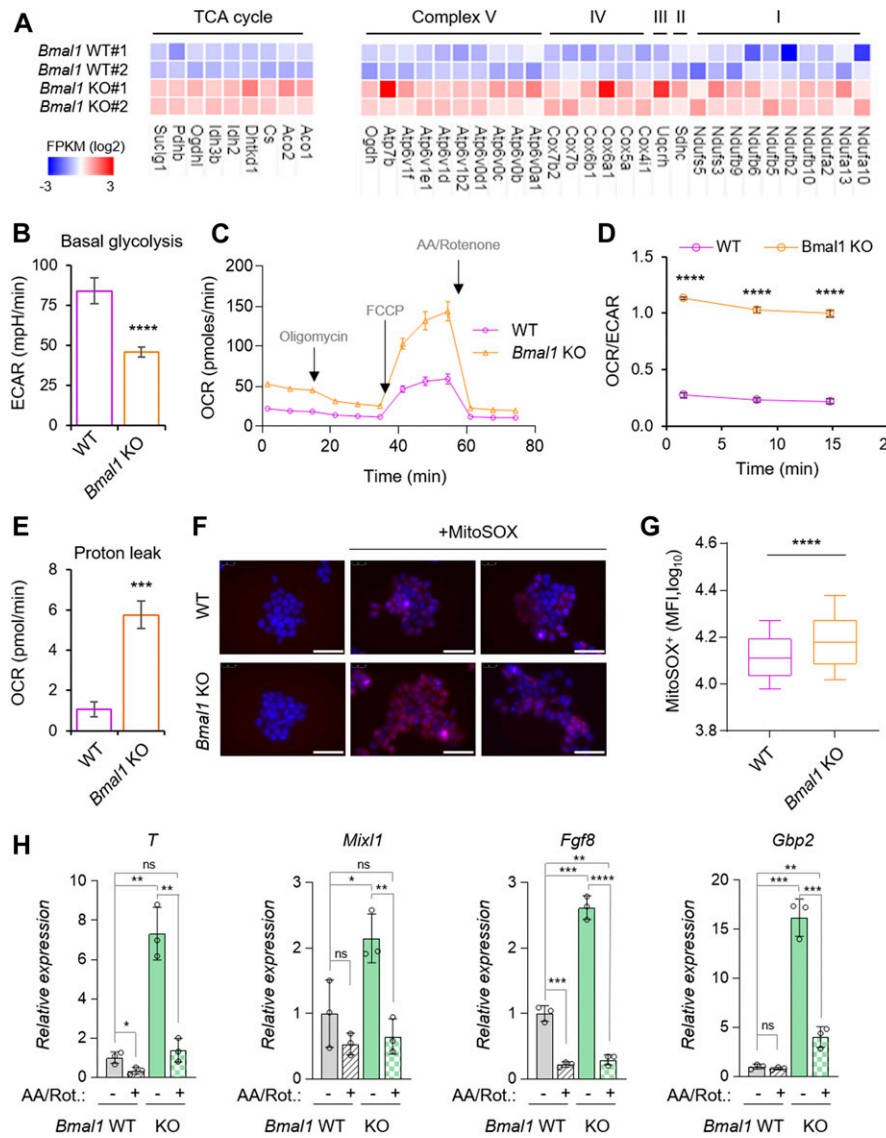

**Figure 7. Depletion of *Bmal1* alters metabolic gene pathways, causing an increase in respiration and mitochondrial ROS production in embryonic stem cells (ESCs).**

**(A)** Heat map of the expression of tricarboxylic acid cycle (TCA) and mitochondrial complex I–V genes from wild-type (*Bmal1* WT) and *Bmal1* KO ESCs RNA-seq. **(B)** Basal glycolysis levels determined as extracellular acidification rate (ECAR) in wild-type (WT) and *Bmal1* KO ESCs (n = 3). **(C)** Bioenergetics assays in ESCs from WT and *Bmal1* KO ESCs. The different drugs used in the assay are indicated in grey (n = 3). **(D)** OCR/ECAR under basal conditions for the indicated cell lines at three different time points (n = 3). **(E)** Proton leak shown as OCR in WT and *Bmal1* KO ESCs (n = 3). **(F)** Fluorescence representative microscopy images showing MitoSOX fluorescence (red) in WT and *Bmal1* KO ESCs. Nuclei were stained with DAPI and shown in blue. The scale bar represents 50 μm. **(G)** Box plots showing the mean fluorescence intensity of MitoSOX treated *Bmal1* WT and KO cells. Center lines, medians; whiskers extend to the 10th/90th percentile. **(H)** Relative expression of the indicated genes in *Bmal1* WT and KO cells with or without antimycin A/rotenone (AA/Rot) treatment. *t* test was performed. (n = 3). In (B, C, D, E, H), data are represented as mean ± SD. Two-tailed unpaired *t* test was performed. ****$P < 0.0001$, ***$P < 0.001$, **$P < 0.01$, *$P < 0.05$; AA, Antimycin A; FCCP, carbonyl cyanide-4 phenylhydrazone; ns, not significant; OCR, oxygen consumption rate.

Source data are available for this figure.

re-enter the mitochondrial matrix with no contribution to ATP generation, in a process known as mitochondrial uncoupling or proton leak (Cadenas, 2018). Indeed, we observed that the increase of basal oxidative metabolism in cells lacking BMAL1 (Fig S4D) was accompanied by an increase in proton leak (Fig 7E) and a reduction in the coupling efficiency of mitochondria (Fig S4E). Although the exact link between proton leak and mtROS production is controversial, it is now clear that there is a mutual connection between these two processes (Nanayakkara et al, 2019). To interrogate the effect of *Bmal1* depletion on the production of mtROS species, we stained WT and *Bmal1* KO ESCs with MitoSOX Red. Interestingly, we observed that ESCs lacking BMAL1 displayed a significant increase in mtROS levels compared with WT cells (Figs 7F and G and S4F). Collectively, these findings show that BMAL1 is required for proper metabolic dynamics and mitochondrial function of pluripotent ESCs. Taking into account that the balance between glycolysis and OXPHOS is critical for

modulating the differentiation potential of pluripotent cells (Wu et al, 2016; Cliff & Dalton, 2017; Zhang et al, 2018; Dahan et al, 2019), we tested whether reducing OXPHOS activity in *Bmal1* KO ESCs could restore the proper expression of lineage specification markers during in vitro differentiation. For this purpose, we used an early differentiation in vitro assay to rapidly induce the expression of genes involved in mesendoderm (ME) lineage choice (Thomson et al, 2011). In line with our previous differentiation assays, absence of BMAL1 significantly alter the expression of lineage markers upon ME differentiation (Fig 7H). Importantly, presence of mitochondrial respiration inhibitors (i.e., antimycin A/rotenone [AA/Rotenone]) during ME differentiation of *Bmal1* KO ESCs was sufficient to significantly rescue the expression of *T*, *Mixl1*, *Fgf8*, and *Gpb2* lineage markers compared with *Bmal1* WT ESCs (Fig 7H). These data further suggest that BMAL1 influences early cell fate specification during in vitro differentiation through the modulation of OXPHOS activity in ESCs.

# Discussion

Here, we demonstrate that BMAL1 is not required for ESC maintenance. In contrast, we show that this core component of the circadian clock plays critical roles during ESC in vitro differentiation. Indeed, depletion of *Bmal1* caused the deregulation of genes from the three germ layers (i.e., ectoderm, mesoderm, and endoderm). In line with this, we show that the absence of BMAL1 affects EB formation and gastrulation, using two independent differentiation in vitro organoid systems. Similar conclusions were drawn from an independently performed study, which recently came to our attention (Gallardo et al, 2020). Mechanistically, we find that depletion of *Bmal1* induces a shift in the basal metabolism of pluripotent cells involving an increase OXPHOS activity accompanied by augmented mtROS species production. Importantly, decreasing OXPHOS activity is sufficient to restore proper induction of lineage-specific marker expression during mesendoderm differentiation in the absence of BMAL1.

For circadian oscillation to occur, CLOCK and BMAL1 need to heterodimerize to activate their downstream transcriptional targets, which include *Per1/2/3* and *Cry1/2* (Takahashi, 2017). In agreement with the observation that circadian transcriptional oscillations are not functional in ESCs (Kowalska et al, 2010; Yagita et al, 2010; Umemura et al, 2014, 2017; Dierickx et al, 2017), when we analysed *Cry1/2* and *Per1/2/3* expression in *Bmal1* KO cells, we did not observe any significant change in their transcriptional levels (Table S1), further supporting a clock-independent function for BMAL1 in pluripotent stem cells. Several reports have demonstrated that circadian oscillations are absent from pluripotent cells by using bioluminescent reporter systems and analysing of clock gene expression (Kowalska et al, 2010; Yagita et al, 2010; Umemura et al, 2014, 2017; Dierickx et al, 2017). Nevertheless, both BMAL1 and CLOCK proteins have been reported to be present in ESCs by us (this study) and others (Lu et al, 2016), respectively, posing the question of how circadian oscillations are inhibited in pluripotent stem cells. One possible explanation was indicated by a study that demonstrated that in ESCs, PER is retained in the cytoplasm, therefore avoiding the proper nuclear function of the negative feedback loop required for cyclic circadian regulation (Umemura et al, 2014). Moreover, in contrast to Lu and colleagues (Lu et al, 2016), a recent study claimed that despite being expressed at the mRNA level, CLOCK protein was absent in ESCs because of microRNA-mediated posttranscriptional repression (Umemura et al, 2017). Although the absence of transcriptional cycling in pluripotent cells seems faithfully demonstrated, the extent to which each of the proposed mechanisms contributes to inhibit the normal function of the clock remains to be clarified. Importantly, CLOCK and BMAL1 seem to have opposite non-overlapping functions in terms of ESC differentiation (Gingold et al, 2014). This is in agreement with our observation that Bmal1 depletion facilitated the down-regulation of the pluripotency factor *Nanog* in conditions of mild differentiation, whereas Clock appeared to have the opposite effect, in agreement with what Lu and colleagues showed previously (Lu et al, 2016). Future studies are required to scrutinize common and specific roles of CLOCK and BMAL1 in ESCs, and additional levels of regulation that can shed light into the mechanisms by which pluripotent cells lack functional circadian rhythms.

Previous reports on the effect of loss of *Bmal1* expression on early embryo development are not consistent. Whereas Bradfield et al reported a normal mendelian ratio of *Bmal1$^{-/-}$* pups being born from the mating of two heterozygous *Bmal1$^{+/-}$* mice (Bunger et al, 2000), several recent reports have shown that *Bmal1* KO zygotes show reduced blastocyst formation and post-implantation development (Xu et al, 2016, 2017). Importantly, these defects could be directly associated with intrinsic developmental issues, given that they used females with at least one *Bmal1* allele, which have been reported to possess normal reproduction physiology (Xu et al, 2017). Our observation that depletion of *Bmal1* in ESCs, which are equivalent to the inner cell mass of the preimplantation blastocyst (around E3.5), show defects during in vitro differentiation to the three germ layers (i.e., ectoderm, endoderm, and mesoderm) would further support an embryo-autonomous role for BMAL1 in embryonic development. Taking into consideration the sterility issues described both in male and female *Bmal1* KO mice (Alvarez et al, 2008; Boden et al, 2010; Ratajczak et al, 2012; Liu et al, 2014; Xu et al, 2016), the in vivo study of *Bmal1* KO embryos is limited by the need of crossing heterozygous progenitors. We propose that ESC differentiation through gastruloid aggregation represents a powerful platform to investigate BMAL1 function in early stages of development where circadian rhythms have not yet been established. Moreover, this system allows to differentiate embryo-autonomous circadian mechanisms, excluding the influence from central and other periphery circadian clocks, which are also affected by the disruption of BMAL1 function.

In this study, we demonstrate for the first time a non-canonical function of BMAL1 in cells lacking circadian oscillations. This is in line with the observation that conditional depletion of *Bmal1* at the adult age does not recapitulate many of the metabolic and age-related pathologies observed in *Bmal1* KO mice (Yang et al, 2016). This observation implies that BMAL1 may have several important circadian-dependent and circadian-independent functions during embryogenesis. Importantly, the higher age-dependent accumulation of reactive oxygen species (ROS) observed in several tissues of *Bmal1* KO animals (Kondratov et al, 2006) could explain the early onset of age-associated pathologies observed in these animals. In line with this, we observed an increase in the mtROS levels in ESCs lacking *Bmal1*, even when the clock is not yet active, pointing to a possible factor contributing to the aging phenotypes observed in these animals later during their lifetime. Moreover, previous studies have demonstrated that oxidative stress negatively impacts oocyte quality and fertilization, together with embryo development (Matsuzuka et al, 2005; Tamura et al, 2008). Thus, the decrease in these parameters observed in *Bmal1$^{-/-}$* in vivo may not only be caused by the reproductive female organs but also from excess of ROS at the embryo level.

Finally, we observed that ablation of *Bmal1* in ESCs affected their metabolism by reducing glycolysis and increasing oxidative phosphorylation (OXPHOS). In line with this observation, using genome-wide immunoprecipitation-based techniques, BMAL1 has been shown to target genes related to cellular metabolism in somatic cells (Hatanaka et al, 2010; Wu et al, 2017; Reinke & Asher, 2019), therefore pointing to a potential direct regulation of metabolic genes by BMAL1 through chromatin binding. Although until recently changes in metabolism were thought to be a consequence

of differentiation, it is now clear that metabolism plays an active role in cell fate commitment, mainly by modifying the epigenome, which can in turn regulate pluripotency, differentiation, and somatic cell reprogramming (Dahan et al, 2019). Pluripotent stem cells favour glycolysis over OXPHOS, thus both reducing potential DNA damage by ROS, whereas increasing the amount of available intermediate metabolites for biosynthesis of lipids and nucleotides. Differentiation of ESCs shows a switch towards a more oxidative metabolic status, and interfering with glycolysis or OXPHOS processes has been shown to interfere with lineage commitment (Teslaa & Teitell, 2015; Cliff & Dalton, 2017). In line with this, we observed that BMAL1 is not only required for proper metabolic gene expression but also for appropriate induction of differentiation by EB or gastruloid formation. In agreement with this observation, we observed that global gene expression profile induced by *Bmal1* depletion was significantly similar to those of loss-of-function of pluripotent factors involved in metabolic regulation such as POU5F1, ZFX, or PRC2 members (i.e., EED and SUZ12) (Galan-Caridad et al, 2007; Chen et al, 2008; Brookes et al, 2012; Di Croce & Helin, 2013; Kim et al, 2015). Future studies will be needed to dissect to what extent BMAL1 metabolic targets at the chromatin level are overlapping or different between pluripotent and somatic cells. Intriguingly, ESCs have been shown to present an oscillatory behaviour in glucose transporter expression (Paulose et al, 2012) even before the onset of a functional circadian clock. These data, together with our study, open up a new exciting field to interrogate circadian-independent BMAL1-mediated cyclic regulation of metabolism.

In sum, this novel function of BMAL1 in pluripotency exiting through metabolic regulation opens new avenues for exploring the involvement of non-canonical circadian clock regulation in development and disease.

# Materials and Methods

## Cell lines and cell culture

ESCs were cultured on 0.1% gelatin-coated plates at 37°C with 5% $CO_2$ and maintained in DMEM (high glucose, D6429; Sigma-Aldrich), containing 15% FBS (10270-106; Gibco), 2 mM L-glutamine (G7513; Sigma-Aldrich), 1× nonessential amino acids (11140; Gibco), 1% nucleoside mix, 1× penicillin/streptomycin (15140; Gibco), $10^{-4}$M 2-mercaptoethanol (M6250; Sigma-Aldrich), and 1,000 U/ml of recombinant LIF.

HEK293T cells were cultured at 37°C with 5% $CO_2$ in DMEM (high glucose, D6429; Sigma-Aldrich), containing 10% FBS (10270-106; Gibco), 2 mM L-glutamine (SH30034.01; GE Healthcare), 1× penicillin/streptomycin (15140; Gibco), and $10^{-4}$ M 2-mercaptoethanol (M6250; Sigma-Aldrich).

## RNA extraction

Total RNA was extracted using the E.Z.N.A. Total RNA Kit I kit (R6834-02; Omega Bio-Tek) following the manufacturer's instructions.

For RNA sequencing (RNA-seq), the RNA samples were extracted using Trizol (15596018; Invitrogen) and Phasemaker Tubes (A33248; Invitrogen) following the manufacturer's instructions.

## RT-qPCR analysis

RNA was converted into cDNA using qSCRIPT (84034; Quanta). Quantitative PCR was performed using the PowerUp SYBR Green Master Mix (4367659; Thermo Fisher Scientific) on the StepOnePlus Real-Time PCR System (Applied Biosystem). Gene-specific primers used are provided in Table S2.

## ShRNA design, lentivirus production, and cell transduction

shRNA sequences specific for mouse transcripts were found in the GPP Web Portal Tool available at https://portals.broadinstitute.org/gpp/public/. Oligos targeting the Luciferase and Bmal1 transcripts were cloned in a lentiviral pLKO.1 puro plasmid. ShRNA-targeted sequences used in this study are shown in Table S2. pLKO.1 puro was a gift from Bob Weinberg (plasmid #8453; Addgene; http://n2t.net/addgene:8453; RRID: Addgene 8453) (Stewart et al, 2003).

For lentiviral production, eight million HEK293T cells were seeded in 150-mm plates and transfected 24 h later with 20 µg of the corresponding pLKO.shRNA plasmid together with 10 µg of psPAX2 and pMD2.G packaging mix using PEI (Polyethylenimine, 408727; Sigma-Aldrich) following the manufacturer's instructions. The next day, fresh medium was added to the cells, and viral supernatants were collected 48 and 72 h after transfection. Viruses were then concentrated using centrifugal filter units with 0.22-µm pore size (UFC903024; Amicon).

## AP staining

The presence of AP enzyme is characteristic of undifferentiated cells (Stefkova et al, 2015) and is, therefore, routinely used as a pluripotency marker. For AP staining, plates containing ESCs cultured in the presence or absence of LIF were washed with PBS with $Ca^{+2}$ and $Mg^{+2}$ (Dulbecco's Phosphate Buffered Saline, D8662; Sigma-Aldrich), and the staining was performed according to the instructions provided by the manufacturer (86R-1KT; Sigma-Aldrich). The plates were air-dried and were kept at RT.

## Colony formation assay

To analyse the capability to form colonies from single cells of knockdown and KO lines, colony formation assays were performed as described previously (Fidalgo et al, 2016) with some modifications. Briefly, 800 cells were seeded in six-well plates coated with 0.1% gelatin and cultured in ESCs standard medium. 24 h later, the medium was changed to normal ESC medium or ESC medium without LIF. The media were changed every other day onwards. After 4 d, AP staining was performed as described above. The number of colonies was counted and classified according to their pluripotent state: undifferentiated (AP positive), partially differentiated (partially AP positive), or differentiated (AP negative).

## Western blot analysis

Whole cell extracts were obtained by lysing ESCs subjected to the different treatments with RIPA Lysis Buffer System (sc-24948; Santa Cruz Biotechnology) with freshly added protease and phosphatase

inhibitors (sc-24948; Santa Cruz Biotechnology). Lysates were incubated during 30 min at 4°C and centrifuged for 10 min at 10,000$g$ at 4°C to remove cellular debris. Samples were kept at –80°C until use.

Protein samples for Western blot analysis were prepared by adding Laemmli buffer and denaturing for 5 min at 95°C and loaded for resolution in a Novex WedgeWell 4–20% tris-Glycine Gel (XP04205BOX; Invitrogen). For Western blotting, the following antibodies and dilutions were used: $\alpha$BMAL1 (1:2,000, ab93806; Abcam), $\alpha$POU5F1 (1:4,000, sc-5279; Santa Cruz Biotechnology), $\alpha$NANOG (1:2,000, A300-397A-2; Bethyl Laboratories), $\alpha$ZFP42 (1:1,000, sc-514643; Santa Cruz Biotechnology), $\alpha$EZH2 (1:5,000, #5246S; Cell Signaling Technology), $\alpha$SUZ12 (1:1,000, sc-271325; Santa Cruz Biotechnology), and $\alpha$-ACTIN (1:2,000, sc-47778; Santa Cruz Biotechnology) or $\beta$-TUBULIN (1:2,500, sc-55529; Santa Cruz Biotechnology) as loading controls.

## Immunofluorescence

12,000 cells were seeded in wells of 48-well plates coated with 0.1% gelatin and cultured in ESCs standard medium. After 72 h, cells were fixed with 4% paraformaldehyde for 15 min in darkness at RT and washed two times with PBS (Dulbecco's Phosphate Buffered Saline, D8537; Sigma-Aldrich). Then, cells were permeabilized with 0.25% Triton X-100 (T8787; Sigma-Aldrich) diluted in PBS for 5 min at RT followed by two times of PBS washes and blocking with 10% BSA (BP9702-100; Thermo Fisher Scientific) diluted in PBS for 30 min at 37°C. The antibodies $\alpha$POU5F1 (1:1,500, sc-5279; Santa Cruz Biotechnology), $\alpha$SSEA1-488 (1:1,000, MA1-022-D488; Thermo Fisher Scientific), $\alpha$ZFP281 (1:1,000, sc-166933; Santa Cruz Biotechnology), $\alpha$NANOG (1:500, sc-374103; Santa Cruz Biotechnology), and $\alpha$SOX2 (1:1,000, sc-365823; Santa Cruz Biotechnology) were diluted in 3% BSA and incubated overnight in darkness at 4°C. To detect non-labelled primary antibodies, a *rhodamine red-x* secondary antibody was used (715-295-151; Jackson ImmunoResearch) for 1 h in darkness at RT. Nuclei were stained with DAPI (4',6-diamidino-2-phenilindole, D9542; Sigma-Aldrich).

## Microscopy

Immunofluorescence images were taken with a Leica DMI 6000 inverted microscope with a 40× or 20× objective. The image processing was performed using the Leica Application Suite X, v3.0.11.20652. Bright-field and colour images were taken with a camera Olympus DP72 coupled to an inverted Olympus IX51 microscope.

## Genomic edition using CRISPR/Cas9 technology

*Bmal1* KO ESCs were generated with CRISPR/Cas9 editing tool as previously described (Ran et al, 2013). Briefly, an sgRNA targeting the exon 5 of the *Bmal1* gene was designed using the webtool available at http://crispr.mit.edu and was cloned into the pSpCas9(BB)-2A-Puro (PX459) V2.0 vector. After verifying the correct sequence by SANGER sequencing, ESCs were transfected with the sgRNA-containing plasmid using the jetPRIME kit (114-07; Polyplus) following the manufacturer's instructions. Transfected cells were seeded at

clonal density and selected with 2 $\mu$g/ml puromycin to obtain single colonies suitable for manual picking. Several clones were isolated, and the identity of a KO cell line was confirmed by Sanger sequencing and Western blot analysis. pSpCas9(BB)-2A-Puro (PX459) V2.0 was a gift from Feng Zhang (plasmid #62988; Addgene; http://n2t.net/addgene:62988; RRID: Addgene 62988) (Ran et al, 2013).

## DNA extraction and genotyping

Cells were resuspended in Cell Lysis Solution (158908; QIAGEN) and treated with RNAse A (EN0531; Thermo Fisher Scientific) for 1 h. Then, Protein Precipitation Solution (158912; QIAGEN) was added and samples were centrifuged at 16,000$g$ for 10 min. Once the samples were clean of protein, the DNA was precipitated using isopropanol and washed with ethanol 70%. Finally, genomic DNA was resuspended in DNA Hydration Solution (158916; QIAGEN) overnight at RT.

A three-primer strategy was designed to detect homozygotic deletions in the CRISPR/Cas9–transfected clones as schematized in Fig S2A. Oligo sequences are shown in Table S2.

## RA differentiation assay

For this assay, NG4 ESCs containing a GFP reporter under the control of the pluripotency-related *Nanog* promoter were used. 50,000 cells transfected with shRNA against Bmal1 or Luciferase as a control were seeded in 12-well plates coated with 0.1% gelatine and cultured at 37°C with 5% $CO_2$ in ESCs standard medium. After 24 h, the medium was maintained as control or switched for standard medium without LIF containing 10 nM of RA. After 48 h in RA-containing medium, the cells were analysed by flow cytometry in a BD Accuri C6 Plus instrument (BD Biosciences).

## EB differentiation assay

EBs were formed by aggregation of ESCs in suspension in a low-attachment petri dish as previously described (Doetschman et al, 1985). Briefly, ESCs transfected with shRNA against Bmal1 and Luciferase as control were seeded at 155,000 cells/ml in low-attachment plates and cultured at 37°C with 5% $CO_2$ in standard ESC medium without LIF. The medium was refreshed every other day and samples were collected at day 6 of EB formation.

## Gastruloid aggregation assay

Gastruloid aggregation assays were performed as previously described (Baillie-Johnson et al, 2015; Beccari et al, 2018). Briefly, for the generation of aggregates, an average of 400 ESCs were resuspended in 40 $\mu$l of N2B27 medium (50% DMEM/F12 [Nutrient Mixture F-12 Ham, D6421; Sigma-Aldrich], 50% Neurobasal Medium [21103-049; Gibco] containing 1× B27 and N2 supplements [17504-044 and 17502-048, respectively; Gibco], 2 mM L-glutamine [SH30034.01; GE Healthcare], 1× penicillin/streptomycin [15140; Gibco]), seeded in non–tissue-culture treated u-bottomed 96-well plates and cultured at 37°C with 5% $CO_2$. 48 h after seeding, a 3-$\mu$M CHIR stimuli was applied for 24 h, with media being changed every day. At 120 h, pictures were taken and 16 gastruloids were pooled in 200 $\mu$l of Trizol for RNA extraction as detailed above.

To calculate *Bmal1* WT and *Bmal1* KO gastruloid generation efficiency, the percentage of positive gastruloid-like structures in each experiment was compared (n = 14 groups of eight aggregates each). Gastruloid size measurement was performed using ImageJ Software, quantifying 112 gastruloids per condition.

### Mesendoderm differentiation assay

Mesendoderm differentiation assay was performed as previously described (Thomson et al, 2011). Briefly, *Bmal1* KO or WT ESCs were maintained for a couple of passages in N2B27 supplemented with LIF, 10 ng/ml BMP4 (11330852; Gibco), and 1 $\mu$M MEK inhibitor PD 0325901 (PZ0162; Sigma-Aldrich). Then, 1.5 × 10$^4$ cells/cm$^2$ were seeded on gelatin-coated plates in N2B27 medium. 48 h later, the medium was switched to N2B27 supplemented with CHIR99021 3 $\mu$M (SML1046; Sigma-Aldrich) to induce mesendoderm differentiation. To diminish OXPHOS activity, the cells were treated with rotenone and antimycin A at a concentration of 50 nmol/l each. Rotenone is an inhibitor of the electron transport chain (ETC) complex I, and antimycin A is a complex III inhibitor. After 36 h, total RNA was extracted using the E.Z.N.A. Total RNA Kit I kit as described above.

### Teratoma formation

For teratoma formation, all animal procedures were performed in accordance with the University of Extremadura's Institutional Animal Care and Use Committee. Eight female Swiss nude mice (Charles River Laboratory) at 5 wk of age were used for this assay. *Bmal1* WT and *Bmal1* KO ESCs were grown and passaged at 80% confluence. At the time of injection, the cells were washed with PBS-EDTA, trypsinized, and counted. Aliquots of 200 $\mu$l of 1:4 Matrigel/PBS containing 1 × 10$^6$ *Bmal1* WT or *Bmal1* KO cells were injected subcutaneously in each mouse flank. Teratomas were excised 3–4 wk after injection, measured, and processed for teratoma biopsy histology. Tumours were fixed overnight in formalin, embedded in paraffin, sectioned at 5 $\mu$m, and stained with haematoxylin and eosin. Histological evaluation was performed using a Nikon TE2000-U microscope and ACT-1 Software.

### RNA-seq generation and analysis

RNA extraction method for RNA-seq samples is described above. Reads from the sequencing of biological replicates of WT and *Bmal1* KO ESCs were aligned to the mouse genome (GRCm38, mm10) using TopHat (v2.1.1) and Bowtie2 (v2.3.0) with the default parameters settings, providing a GTF containing known transcripts (-G). Transcript assembly and differential expression analysis were performed by Cufflinks (v2.2.1), expression of transcripts sharing each gene_id was quantified as FPKM (fragments per million mapped reads), and significance of differential expression tests was determined with the Benjamini–Hochberg correction for multiple testing. Differentially expressed genes were identified as genes having more than twofold change in expression in *Bmal1* KO compared with WT cells and an FDR value of less than 0.05.

### GO, KEGG pathway and Network2Canvas analyses

GO and KEGG analyses were performed with DAVID functional annotation tool at http://david.abcc.ncifcrf.gov/tools.jsp (Huang da et al, 2009) with a reference list including all *Mus musculus* genes from NCBI. Network2Canvas (Network Visualization on a Canvas with Enrichment Analysis) (http://www.maayanlab.net/N2C/) (Tan et al, 2013) was used to interrogate the correlation of gene expression changes in *Bmal1* KO ESCs with transcriptional changes upon loss-of-function of transcription regulators from published datasets.

### GSEA

GSEA (v3.0, accessible at http://software.broadinstitute.org/gsea/index.jsp) was used to analyse the enrichment differences between *Bmal1* WT and KO cells. Briefly, the analysis was performed running with 1,000 permutations and gene_set permutation type. *t* test metric to rank genes and weighted_p2 for enrichment statistics were used. Enrichment plot, normalized enrichment score, statistical significance (*P*-value), and FDR were calculated by the software.

### Extracellular metabolic flux analysis

45,000 *Bmal1* WT and KO ESCs were seeded in Seahorse XFp Cell Culture Miniplates (103025-100; Agilent) coated with 0.1% gelatin and cultured at 37°C with 5% $CO_2$ in standard ESCs medium for 24 h. OCR and extracellular acidification rate were measured using Seahorse XFp Cell Mito Stress Test Kit (103010-100; Agilent) and Seahorse XFp Glycolitic Rate Assay Kit (103346-100; Agilent), respectively, following the manufacturer's instructions. Briefly, the cells were incubated 45–60 min in Seahorse XF DMEM Medium, pH 7.4 (103575-100; Agilent), supplemented with pyruvate 1 mM, L-glutamine 2 mM, glucose 10 mM, and Hepes 5 mM (this only for the Glycolytic Rate Assay), and mitochondrial function was analysed in Seahorse XFp Analyser (Agilent).

Proton leak and coupling efficiency parameters were calculated using the OCRs data obtained with the Seahorse XFp Cell Mito Stress Test Kit (103010-100; Agilent). The assay involves the use of three different drugs: oligomycin, which decreases the electron flow through the ETC, resulting in a mitochondrial respiration reduction; carbonyl cyanide-4 phenylhydrazone (FCCP), which restores the electron flow through the ETC obtaining the maximal oxygen consumption; and AA/Rotenone, which completely blocks mitochondrial respiration and allows the calculation of non-mitochondrial oxygen consumption. Proton leak was calculated as the minimum OCR measurement after oligomycin injection minus the non-mitochondrial oxygen consumption, determined as the median of the OCR data after AA/Rotenone addition. Coupling efficiency was calculated as the ATP-linked OCR divided by the basal mitochondrial OCR and multiplied by 100.

### mtROS analysis

The same number of *Bmal1* WT and KO ESCs was seeded in 12-well plates in ESC standard medium. 24 h later, the cells were stained with MitoSOX Red mitochondrial superoxide indicator for live-cell

imaging (M36008; Invitrogen) following the manufacturer's instructions. Briefly, the samples used for cytometry analysis were dissociated with trypsin (SH30236.01; GE Healthcare) and washed with PBS with Ca$^{+2}$ and Mg$^{+2}$ (Dulbecco's Phosphate Buffered Saline, D8662; Sigma-Aldrich). Then, the cells were incubated for 10 min in 5 $\mu$M MitoSOX reagent working solution in PBS or only in PBS as a control and analysed by flow cytometry in a BD Accuri C6 Plus instrument (BD Biosciences).

Cells used for imaging were treated with 5 $\mu$M MitoSOX reagent working solution in PBS on plates for 10 min at 37°C. Then, nuclei were stained with DAPI, and images were acquired with a Leica DMI 6000 inverted microscope with a 20× objective.

## Statistical analysis

Differences with two-sided $P < 0.05$ were deemed statistically significant, as evaluated using the $t$ test. Results are reported as mean ± SD or SEM as indicated in the figure legends.

# Data Availability

RNA-seq datasets generated in this study are deposited at Gene Expression Omnibus (accession number GSE133908).

# Supplementary Information

# Acknowledgements

This research was funded by Spanish Agencia Estatal de Investigación and co-funded by the FEDER Program of the European Union (BFU2016-80899-P to M Fidalgo and RTI2018-096708-J-I00 to D Guallar) (AEI/FEDER, EU), the Xunta de Galicia-Consellería de Cultura, Educación e Ordenación Universitaria (ED431F 2016/016 to M Fidalgo), and the Fundación Ramón Areces (2016-PO025 to M Fidalgo). This work was also supported by BFU2017-85547-P grant from the Ministry of Economy, IB18014 and GR15164 from Junta de Extremadura to JM Carvajal-Gonzalez. M Fidalgo and JM Carvajal-Gonzalez are recipients of Ramón y Cajal awards (RYC-2014-16779 and RYC-2015-17867, respectively) from the Ministry of Economy and Competitiveness (MINECO) of Spain. A Fuentes-Iglesias, V Garcia-Outeiral, and A Escudero are recipients of fellowships (MINECO, BES-2017-082007 to A Fuentes-Iglesias and Ministerio de Ciencia, Innovación y Universidades; FPU17/01131 to V Garcia-Outeiral and FPU2018/01246 to A Escudero).

## Author Contributions

C Ameneiro: conceptualization, resources, formal analysis, validation, investigation, visualization, methodology, and writing—original draft.

T Moreira: resources, formal analysis, validation, investigation, and methodology.

A Fuentes-Iglesias: resources, data curation, software, formal analysis, validation, investigation, and visualization.

A Coego: formal analysis, validation, and investigation.

V Garcia-Outeiral: formal analysis, validation, investigation, visualization, and methodology.

A Escudero: formal analysis, validation, investigation, visualization, and methodology.

D Torrecilla: formal analysis, investigation, visualization, and methodology.

S Mulero-Navarro: resources, formal analysis, validation, investigation, visualization, and methodology.

JM Carvajal-Gonzalez: resources, formal analysis, funding acquisition, validation, investigation, visualization, and methodology.

D Guallar: conceptualization, resources, data curation, formal analysis, supervision, funding acquisition, validation, investigation, visualization, methodology, project administration, and writing—original draft, review, and editing.

M Fidalgo: conceptualization, resources, data curation, formal analysis, supervision, funding acquisition, validation, investigation, visualization, methodology, project administration, and writing—original draft, review, and editing.

## Conflict of Interest Statement

The authors declare that they have no conflict of interest.

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
