## [Reviewer comments · Life Science Alliance]

Life Science Alliance

BMAL1 coordinates energy metabolism and differentiation of pluripotent stem cells

Cristina Ameneiro, Tiago Moreira, Alejandro Fuentes-Iglesias, Alba Coego, Vera Garcia-Outeiral, Adriana Escudero, Daniel Torrecilla, Sonia Mulero-Navarro, Jose Carvajal-Gonzalez, Diana Guallar, and Miguel Fidalgo

DOI: <https://doi.org/10.26508/lsa.201900534>

Corresponding author(s): Miguel Fidalgo, University of Santiago de Compostela and Diana Guallar, Universidade de Santiago de Compostela

Review Timeline:

Submission Date:	2019-08-23
Editorial Decision:	2019-10-16
Appeal Received	2019-10-25
Editorial Decision	2019-10-30
Revision Received:	2020-03-07
Editorial Decision:	2020-03-23
Revision Received:	2020-03-30
Editorial Decision:	2020-03-31
Revision Received:	2020-03-31
Accepted:	2020-04-01

Scientific Editor: Andrea Leibfried

Transaction Report:

October 16, 2019

Re: Life Science Alliance manuscript #LSA-2019-00534-T

Miguel Fidalgo
University of Santiago de Compostela
Physiology
Barcelona Avenue s/n-Campus Vida
Santiago de Compostela, A Corunha 15782
Swaziland

Dear Dr. Fidalgo,

Thank you for submitting your manuscript entitled "BMAL1 coordinates energy metabolism and differentiation of pluripotent stem cells" to Life Science Alliance. Please excuse the delay in getting back to you, we had to give the reviewers more time in this case. We have now heard back from two reviewers on your work and their reports are attached below.

As you will see, while the reviewers appreciate some aspects of your work, they also think that the data quality is not sufficient for publication here and that your conclusions are not fully supported by the data provided. Importantly, the conclusions on self-renewal, pluripotency and effects on gastrulation are not sufficiently supported. They further note that the observed effects could be due to indirect consequences of loss of Bmal1 and that the insight into altered metabolic gene expression is too limited. Finally, the mismatch with the current knowledge on metabolic state and effects on pluripotency remains unexplained.

Given these concerns, we are afraid we do not have the level of reviewer support that we would need to proceed further with the paper. We are thus returning your manuscript to you with the message that we cannot publish it here.

We are sorry our decision is not more positive, but hope that you find the reviews constructive. Of course, this decision does not imply any lack of interest in your work and we look forward to future submissions from your lab.

Thank you for your interest in Life Science Alliance.

Sincerely,

Andrea Leibfried, PhD
Executive Editor
Life Science Alliance

Reviewer #1 (Comments to the Authors (Required)):

In this manuscript, the authors demonstrated that BMAL1 was dispensable for ESC self-renewal, but played a role in the exit of pluripotency and cell differentiation commitment. Mechanistically, the authors illustrated that BMAL1 participated in regulating energy metabolism and mitochondrial function.

Overall, the part that BMAL1 was involved in the metabolic regulation is interesting. However, the other parts of this paper are not very conclusive due to either lack of statistical analysis or bad quality of the data. More experiments is needed to clarify these issues.

Concerns:

- (1) In Fig1A, the authors examined the RNA level of Bmal1, Clock, Nanog and Pou5f1 in MEFs and ESCs. Since RNA level could not fully represent protein level, western blotting analysis of these proteins and statistical analysis is required.
- (2) The quality of western blotting in Fig2A and Fig 2B are bad. Better quality of western blotting and statistical analysis of Nanog or the other markers should be performed.
- (3) In Fig2A, it should be 22 bp deletion in Bmal1 KO ESCs instead of 20 bp.
- (4) In the text, the authors claimed that the self-renewal of ESCs was not greatly affected by deletion of BMAL1 neither on colony number nor on the colony morphology (Fig 2F and 2G). However, in Fig2G, it seems that knockout of Bmal1 affects the differentiation. Convincing conclusion should be stated. In addition, statistical analysis is missing in Fig 1F.
- (5) ESC pluripotency analysis is required in in vivo experiment to check whether BMAL1 KO ESCs can form the three germ layers.
- (6) In Fig 5D, BMAL1 deficiency resulted in abnormal in vitro gastrulation, which lacks statistical analysis. More evidences are needed to confirm the relationship between aberrant upregulation of three-germ-layer markers and the deficiency of polarization or elongation during in vitro gastrulation.
- (7) The authors mentioned that close correlation of the transcriptional changes of regulators related with cellular differentiation and cellular metabolism including Zfx and Polycomb members (i.e. Eed and Suz12) were found upon depletion of Bmal1 and Pou5f1 (Fig EV3A) . Experiments including RT-qPCR or western blotting of these regulators in Bmal1 KO and Pou5f1 KO cells should be performed to support the conclusion.
- (8) Scale bar is missing in most of the staining data, such as in Fig1C, 1D, 5C, 7H, EV1B.

Reviewer #2 (Comments to the Authors (Required)):

The studies by Ameneiro and colleagues investigate the function of circadian gene BMAL1 in regulating embryonic stem cells (ESC) differentiation and metabolism. They first show that BMAL1 is dispensable for ESC pluripotency. They further show using both RNA interference and CRISPR-Cas9 that loss of BMAL1 does not significantly alter pluripotency in ESCs. Then through a set of experiments they propose that BMAL1 may be involved in early decisions during pluripotency exit. They further provide evidence that loss of BMAL1 impairs ESC differentiation and alters ESC gene expression (upregulation of ectoderm vs downregulation of mesoderm and endoderm genes). They also use in vitro gastruloid formation to show that BMAL1 KO cells do not elongate properly in vitro. Finally, they show that loss of BMAL1 is associated with metabolic alterations. This manuscript address an important question regarding BMAL1 regulation of ESC fate determination. The authors have performed enormous amount of work using a combination of approaches and have produced interesting data. However, their conclusion is some instances is premature or not supported by the data.

My comments are as follow:

(1) Figure 6: please clarify how this figure supports BMAL1 control of transcriptional regulation of ESC differentiation. As the authors have noted loss of BMAL1 results in modulations of cellular status (pluripotency vs differentiation). The transcriptomic changes may be just a reflection of cellular differentiation and not the cause. The data does not support: "Thus, these results show that BMAL1 is involved in transcriptional regulation of developmental processes in ESCs" as stated by the authors.

(2) Figure 3: the results in this figure are not clear. Experiments should be explained in more detail and results flushed out.

(3) Figure 7: BMAL1 KO results in modifications of metabolic gene expression (7A). The authors should clarify the identity of these genes (glycolytic? mitochondrial/TCA etc?). This is important in order to support their next finding. In addition, how do they reconcile reduced glycolysis in BMAL KO ESCs with their differentiation status?

(4) Figure 4E: BMAL1 KO shows upregulation of ectoderm genes associated with downregulation of endoderm, mesoderm genes. Does loss of BMAL1 enhance the ESC differentiation to ectoderm (at the expense of other germ layers)?

Minor:

- The authors should quantify the bands in WB (Figure 1B, 2B etc).
- Figure 2D, it is important to show protein expression.
- Figure 5C-D: please provide quantitative data measuring the size of the gastruloid structure and possibly the elongation/polarization.
- How do gene expressions in Figure 5B relate to that in 5E?
- The authors should explain in detail of how the results from Figure 7F-7G have been generated.

Dear Dr. Leibfried,

Thanks for your kind efforts in handling our manuscript and sending us your thoughtful editorial comments.

After carefully reading through your decision letter as well as reviewers' comments, we took time to reevaluate all our data-sets (including some new data-sets) and were able to put together the rebuttal document (see attached PDF file) with the following highlights:

- 1) We feel that the two reviewers recognized the significance and conceptual novelty of our findings to the field and that they have provided many constructive suggestions for us to improve our manuscript;
- 2) We are in a very strong position to address all their concerns within a reasonable time frame. In fact, we have many data-sets available already to address many of their concerns (please see the rebuttal PDF file). These data-sets are now ready to be incorporated into the revised manuscript.
- 3) We have a solid and realistic plan to address the remaining issues raised by both reviewers (please see our rebuttal PDF file).
- 4) We feel that some of the concerns raised by the reviewers are caused by misunderstandings with another manuscript (i.e. data quality not sufficient and conclusions not well supported).

I sincerely hope that you will also find our rebuttal reasonable and our experimental plan realistic and promising to improve our story towards a final product (i.e. a revised manuscript) that meets the publication standards of Life Science Alliance.

Thanks again for your time and effort. Looking forward to hearing from you soon,

Best regards,
Miguel Fidalgo

We appreciate the constructive and in-depth comments of the two reviewers, which have helped us to design and perform additional experiments, thus further clarifying the presentation of our data and strengthening the conclusions of our manuscript. Here we have provide detailed point-by-point responses to reviewers' comments together with our additional information and new experimental data. For the reviewers' convenience, we have highlighted in **red** those new data figures that have been incorporated in the revised manuscript and are also mentioned in this rebuttal.

Reviewer #1 (Comments to the Authors (Required)):

In this manuscript, the authors demonstrated that BMAL1 was dispensable for ESC self-renewal, but played a role in the exit of pluripotency and cell differentiation commitment. Mechanistically, the authors illustrated that BMAL1 participated in regulating energy metabolism and mitochondrial function. Overall, the part that BMAL1 was involved in the metabolic regulation is interesting.

We thank the reviewer for his/her positive comments and appreciation of our work.

However, the other parts of this paper are not very conclusive due to either lack of statistical analysis or bad quality of the data. More experiments are needed to clarify these issues.

We apologize for having forgotten to perform the statistical analysis of some panels in the first version of the manuscript. We have now fixed this in the revised current version of the manuscript. On the other hand, we respectfully disagree with the reviewer on the comment that "*bad quality of the data*" and we think this may be due to a confusion (please see below point no. #1.2).

Concerns:

(1.1) In Fig1A, the authors examined the RNA level of Bmal1, Clock, Nanog and Pou5f1 in MEFs and ESCs. Since RNA level could not fully represent protein level, western blotting analysis of these proteins and statistical analysis is required.

This is a good suggestion from the reviewer. Indeed, we have followed his/her indications and analysed the protein expression levels of BMAL1, NANOG and POU5F1 (**Additional Figure 1A**) (**Figure 1A**). Importantly, consistent with the RNA expression (**Figure 1A** in original manuscript version) we found that BMAL1 protein levels are also significantly higher in pluripotent ESCs than in somatic MEFs (**Additional Figure 1B**) (**Figure 1A**).

Additional Figure 1. (A) Western blot of BMAL1, NANOG and POU5F1 in mouse embryonic fibroblasts (MEFs) and embryonic stem cells (ESCs). βTUBULIN was used as loading control. Biological triplicates are shown. Quantification of the blots is shown in **(B)**. Bars represent mean ± s.d. * $P < 0.05$, *** $P < 0.001$, **** $P < 0.0001$ significantly differences calculated using two-tailed unpaired Student's t -test analysis.

(1.2) The quality of western blotting in Fig2A and Fig 2B are bad. Better quality of western blotting and statistical analysis of Nanog or the other markers should be performed.

We respectfully disagree with the reviewer and as stated above, we think that there is a confusion when he/she states that "the quality of western blotting in Fig2A and Fig 2B are bad" because:

- i) **Figure 2A** in the previous version of the manuscript was not a Western Blot (WB), but instead it showed a schematic representation for the CRISPR/Cas9 strategy used in our study.
- ii) The WB showed in **Figure 2B** in the previous version showed unique and clear bands for each of the antibody used (i.e. BMAL1, NANOG, POU5F1, ACTB).

Nevertheless, following the reviewer's suggestion we have now provided new data that replaces **Figure 2B** from the previous manuscript version, with biological triplicates and statistical analysis (**Additional Figure 2**) (**Figure 2B**).

Additional Figure 2. (A) Western Blot of BMAL1 and pluripotent factors ZFP42, NANOG and POU5F1 in *Bmal1* WT and KO ESCs. ACTIN was used as a loading control. **(B)** Quantification of three independent experiments. Bars represent mean ± s.d. ** $P < 0.01$ significantly differences calculated using two-tailed unpaired Student's t -test analysis. ns: not significant.

(1.3) In Fig2A, it should be 22 bp deletion in Bmal1 KO ESCs instead of 20 bp. We apologize that in our current version of the manuscript we mislabeled the deletion as remarked by the reviewer. Now we have fixed this issue (**Additional Figure 3**) (**Figure 2A**).

Additional Figure 3. (Top) Schematic representation for the CRISPR/Cas9 strategy used. The designed sgRNA is underlined and the PAM sequence is highlighted in red. (Bottom) Sequence deleted (22 bp) in *Bmal1* KO alleles detected by SANGER sequencing.

(1.4) In the text, the authors claimed that the self-renewal of ESCs was not greatly affected by deletion of BMAL1 neither on colony number nor on the colony morphology (Fig 2F and 2G). However, in Fig2G, it seems that knockout of Bmal1 affects the differentiation. Convincing conclusion should be stated. In addition, statistical analysis is missing in Fig 1F.

We would like to apologize if our choice of words when we said “was not greatly affected” could generate confusion. Following the reviewer’s recommendation and to avoid any misinterpretation, we have now rephrased it indicating the specific percentages in the text as follows: “On the other hand, when we analysed the self-renewal capability of BMAL1-depleted ESCs we did not observe changes in the number of colonies (Figure 2F) and only a mild reduction in the percentage of undifferentiated colonies (74.5±5.6 versus 59.7±5.7) in presence of LIF (Figure 2G), which is in line with the acute silencing of Bmal1 using the shRNA approach (Figure 1F). Of note, Bmal1 KO ESCs showed a defect in differentiation upon LIF withdrawal as observed by the decrease in the percentage of fully differentiated colonies (65.2±4.2 versus 34.5±3.1) (Figure 2G), suggesting a possible role of BMAL1 during the exit of pluripotency.”

On the other hand, to address his/her suggestion regarding to lack of statistical analysis in **Figure 1F** in the previous version, we have now added that piece of information in the current revised manuscript (**Additional Figure 4**) (**Figure 1F**).

Additional Figure 4. ESCs transfected with shRNA against *Bmal1* or *Luciferase* as control are seeded at low confluence in standard medium with or without LIF, and colonies are counted and classified into the three indicated categories according to their AP staining intensity (n=3). Bars represent mean \pm s.e.m. Significant differences calculated using two-tailed unpaired Student's *t*-test analysis. ns: not significant.

(1.5) ESC pluripotency analysis is required in *in vivo* experiment to check whether BMAL1 KO ESCs can form the three germ layers.

We thank the reviewer for his/her suggestion. Following the reviewer's suggestion we have now analysed the ability of *Bmal1* KO ESCs to form the three layers *in vivo* using teratoma assays. For that purpose, eight female Swiss Nude mice at 5 weeks of age were injected subcutaneously in their flanks with wild type or *Bmal1* KO ESCs. Teratomas were excised 4 weeks after injection, measured and processed for staining with hematoxylin and eosin (H&E). Our results show that *Bmal1* KO ESCs can form teratomas containing differentiated tissues from the three germinal layers (Additional Figure 5) (Figures 4A and 4B). Notably, this data is consistent with our results in the previous manuscript version, which show that *Bmal1* KO ESCs are pluripotent given that: i) BMAL1 is dispensable for the maintenance of the pluripotent cellular state (Figures 1 and 2 in the previous version); ii) even though presence of BMAL1 *in vitro* is required for proper expression of the lineage markers of ectoderm, mesoderm and endoderm germ layers (Figure 4E in the previous version), *Bmal1* KO ESCs are capable of forming embryoid bodies (EBs) (Figures 4C and 4D in the previous version). Altogether, our data support that *Bmal1* KO ESCs are pluripotent and that loss of *Bmal1* leads to defects that interfere with normal expression patterns of lineage specification markers, which are detectable during *in vitro* differentiation. However, we cannot discard the possibility that during *in vivo* differentiation (i.e. by teratoma assays) there are also aberrant expression of specific markers of one or more germ layers even though *Bmal1* KO teratomas seem grossly normal.

Additional Figure 5. (A) Teratomas formed by wild type (WT) and *Bmal1* knock-out (KO) ES cells after 3-4 weeks of injection. (B) Quantification of teratomas size (8 tumors per genotype). Data are shown as mean±s.d. Student's *t*-test was performed. ns: not significant. (C) Histological analysis of teratomas of the indicated genotype by H&E staining. Structures representing the three embryonic germ layers in both genotypes were found. (*): connective tissue. (^): neural tissue and (+): epithelial tissue. Image magnification: 20X.

(1.6) In Fig 5D, BMAL1 deficiency resulted in abnormal *in vitro* gastrulation, which lacks statistical analysis. More evidences are needed to confirm the relationship between aberrant upregulation of three-germ-layer markers and the deficiency of polarization or elongation during *in vitro* gastrulation.

We apologize for not having included the statistical analysis in **Figure 5D** in the previous version of the manuscript. Following the reviewer's suggestion we have now performed additional gastruloid formation assays and included statistical analysis (**Additional Figure 6A**) (**Figure 5E**). Consistent with our previous data, we found that absence of BMAL1 significantly impact in gastruloid formation efficiency compared to wild-type (7.14% vs 41.07%, respectively). Additionally, to further characterize the effect of *Bmal1* depletion in elongation during *in vitro* gastrulation, we have now analysed the size of *Bmal1* WT or KO ESC-derived aggregates during gastruloid formation assay (**Additional Figure 6B**) (**Figure 5D**). Our data supports that absence of BMAL1 impairs the elongation process *in vitro*, giving rise to cellular aggregates displaying reduced size at 120 hours. Likewise, we observed altered transcriptional dynamics of several members of the *Hoxd* gene cluster (**Additional Figure 6C**) (**Figure 5G**), which is one of the hallmarks of axial gene regulatory systems whose sequential activation is associated with the patterning and formation during *in vitro* gastruloid formation by the seminal work of Beccari and collaborators [1]. Thus, these results are in line with the observed deregulation, in absence of BMAL1, of some germ layer markers (**Figures 5E** in the previous version) that have been reported to be timely transcribed during *in vitro* gastruloid formation [1] (i.e. *Sox1* and *Pax3* which should be localized centrally; *Gata4*, *Gata6* and

T/Brachyury expressed in post-occipital structures; *Mixl1* and *Eomes* associated with the gastrulation process *in vivo*).

Additional Figure 6. (A) Percentage of gastruloid-like structures formed by *Bmal1* WT and KO aggregates at 120h of *in vitro* gastrulation. (n=14 independent experiments with at least 8 formed aggregates per experiment). Bars represent mean \pm s.e.m. Two-tailed unpaired Student's *t*-test was calculated. **** $P < 0.0001$. **(B)** Quantification of the size of the aggregates formed by *Bmal1* WT and KO cells at 120h of *in vitro* gastrulation. a.u.: arbitrary units. All data are shown as mean \pm s.e.m. Two-tailed unpaired Student's *t*-test was calculated. **** $P < 0.0001$. **(C)** Expression of *Hoxd1*, *Hoxd3*, *Hoxd4* and *Hoxd10* dynamics during *in vitro* gastrulation of *Bmal1* WT and KO ESCs at the indicated timepoints. (n=3). Data is represented as mean \pm s.e.m.

(1.7) The authors mentioned that close correlation of the transcriptional changes of regulators related with cellular differentiation and cellular metabolism including *Zfx* and Polycomb members (i.e. *Eed* and *Suz12*) were found upon depletion of *Bmal1* and *Pou5f1* (Fig EV3A). Experiments including RT-qPCR or western blotting of these regulators in *Bmal1* KO and *Pou5f1* KO cells should be performed to support the conclusion.

We apologize if in the previous version of the manuscript the message in the “**Results section: *Bmal1* Supports Glycolytic Metabolism in ESCs**” that we tried to convey was not clear enough and led the reviewer to a misunderstanding. What we meant by: “*we found a close correlation of the transcriptional changes upon *Bmal1* depletion with those upon depletion of the master regulator of pluripotency *Pou5f1* [25-27] as well as other pluripotent regulators related with cellular differentiation and cellular metabolism including *Zfx* [28,29] and Polycomb members (i.e. *Eed* and *Suz12*) [30,31] (Fig EV3A)*” was that depletion of *Bmal1*, produces similar transcriptional changes (or at least in part correlate with) to the ones previously reported after depletion of *Pou5f1* or *Zfx* or *Eed* or *Suz12* in mouse ESCs.

Nevertheless, following the reviewer’s suggestion we have now assessed the effect of *Bmal1* depletion in the abundance of *Pou5f1*, *Zfx*, *Eed* or *Suz12* (**Additional Figure 7**) (**Figures EV3C and EV3D**). Our results show that *Bmal1* depletion does not alter the expression of *Pou5f1*, *Zfx*, *Eed* or *Suz12* in mouse ESCs.

In order to avoid any possible misperception or confusion from readers we have now clarified this in the text as follows: “To understand how BMAL1 orchestrates transcriptional programs involved in proper cell differentiation in vitro, we next compared the gene expression changes after *Bmal1* depletion in ESCs with those caused by loss-of-function of other transcription regulators using the Network2Canvas computational tool (see Materials and Methods). Interestingly, we found a close correlation of the transcriptional changes in absence of BMAL1 with those upon depletion of the master pluripotency regulator *Pou5f1* [25-27] as well as other pluripotent regulators related with differentiation and/or cellular metabolism including ZFX [28,29] and Polycomb members (i.e. EED and SUZ12) [30,31] (Figure EV3B). These results, together with the observation that the expression of these factors was not altered upon *Bmal1* depletion in mouse ESCs (Figures EV3C and EV3D), suggest the existence of common pathways regulated by this set of factors and BMAL1.”

Additional Figure 7. (A) RNA expression of the indicated genes in *Bmal1* wild type (WT) and knock-out (KO) ESCs detected by RNA-seq. FPKM: fragments per million mapped reads. **(B)** Western Blot of Polycomb proteins (EZH2 and SUZ12) and the pluripotency master regulator POU5F1 in *Bmal1* WT and KO ESCs. GAPDH was used as a loading control.

(1.8) Scale bar is missing in most of the staining data, such as in Fig1C, 1D, 5C, 7H, EV1B. We apologize as the scale bar size that we used in the previous manuscript version was too small and lead the confusion that they were not present. In order to improve the clarity of the representation, we have now increased the size of the scale bars in all figures across the current manuscript version (**Additional Figure 8**).

Additional Figure 8. Figures numbered as in the previous version of the manuscript are shown. **(Figure 1C)** Bright Field (BF) and Alkaline-phosphatase staining (AP) images of ESCs transfected with shRNA against *Bmal1* or Luciferase as a control. White bars represent 200 μ m. **(Figure 1D)** Immunofluorescence of the pluripotent marker SSEA1 in ESCs transfected with shRNAs against *Bmal1* or Luciferase as control. Nuclei were stained with DAPI. White bars represent 50 μ m. **(Figure 4C)** Representative images of embryoid bodies at day 6 of differentiation in the indicated cell lines. White bars represent 200 μ m. **(Figure 5C)** Images of 120h aggregates for *Bmal1* WT and KO cells. White bars represent 200 μ m. **(Figure 7H)** Fluorescence representative microscopy images showing MitoSOX fluorescence (red) in WT and *Bmal1* KO ESCs. Nuclei were stained with DAPI and shown in blue. White bars represent 50 μ m. **(Figure EV1B)** Immunofluorescence of pluripotent markers POU5F1, and ZFP281 in ESCs transfected with shRNAs against *Bmal1* or Luciferase as control. Nuclei were stained with DAPI. White bars represent 100 μ m.

Reviewer #2 (Comments to the Authors (Required)):

The studies by Ameneiro and colleagues investigate the function of circadian gene BMAL1 in regulating embryonic stem cells (ESC) differentiation and metabolism. They first show that BMAL1 is dispensable for ESC pluripotency. They further show using both RNA interference and CRISPR-Cas9 that loss of BMAL1 does not significantly alter pluripotency in ESCs. Then through a set of experiments they propose that BMAL1 may be involved in early decisions during pluripotency exit. They further provide evidence that loss of BMAL1 impairs ESC differentiation and alters ESC gene expression (upregulation of ectoderm vs downregulation of mesoderm and endoderm genes). They also use in vitro gastruloid formation to show that BMAL1 KO cells do not elongate properly in vitro. Finally, they show that loss of BMAL1 is associated with metabolic alterations. This manuscript address an important question regarding BMAL1 regulation of ESC fate determination. The authors have performed enormous amount of work using a combination of approaches and have produced interesting data. However, their conclusion is some instances is premature or not supported by the data.

We thank the reviewer for these positive comments on our manuscript.

My comments are as follow:

(2.1) Figure 6: please clarify how this figure supports BMAL1 control of transcriptional regulation of ESC differentiation. As the authors have noted loss of BMAL1 results in modulations of cellular status (pluripotency vs differentiation). The transcriptomic changes may be just a reflection of cellular differentiation and not the cause. The data does not support: "Thus, these results show that BMAL1 is involved in transcriptional regulation of developmental processes in ESCs" as stated by the authors.

We appreciate the reviewer's comment and we apologize for the overstatement that "these results show that BMAL1 is involved in transcriptional regulation of developmental processes in ESCs". It is true that while BMAL1 has been widely reported to function as a transcriptional regulator in other cellular contexts (Reviewed in [2]), in this manuscript we have not directly addressed whether BMAL1 regulates in a direct or indirect manner the transcription of the 444 upregulated and 197 downregulated genes (Figure 6A in the previous version). This is an interesting question to be addressed by us or other laboratories in future studies. On the other hand, depletion of *Bmal1* (i.e. KD or KO) in ESCs does not affect:

- a) their self-renewal ability in presence of LIF (**Figures 1G-F and 2F-G** in the previous version)
- b) the expression of key pluripotency gene regulators such as *Pouf51* (also known as *Oct4*), *Nanog*, *Sox2* are the RNA (**Figures EV1A and 2D**) or protein levels (**Figures 1B, 1D, EV1B, 2B, 2E and EV2D** in the previous manuscript version)

Taking into account these data, the transcriptomic changes observed upon *Bmal1* depletion in ESCs in presence of LIF (Figure 6 in the previous version) cannot be just a reflection of the

cellular differentiation status of the cells. Additionally, we have now evaluated whether *Bmal1* depletion in ESCs affects the expression of pluripotency enriched genes [3] as well as the targets of the master regulators of pluripotency POU5F1 (OCT4), SOX2 and NANOG (OSN) [4,5] by gene set enrichment analysis (GSEA) (**Additional Figure 9**) (**Figure EV3A**). Similar to our results mentioned above, absence of BMAL1 does not significantly impact the expression of ESC-enriched genes or OSN targets, further indicating that *Bmal1* KO ESCs remain overall pluripotent. Thus, our results suggest that the transcriptional changes in differentiation-related markers in ESCs in absence of BMAL1 are not sufficient to trigger differentiation under pluripotent cell culture conditions (i.e. presence of LIF). However, transcriptional deregulation of lineage-specific markers in *Bmal1* KO ESCs (**Figures 6B-F** in the previous version) are in line with our data showing that depletion of *Bmal1* greatly affects the *in vitro* differentiation potential of ESCs (**Figures 4 and 5** in the previous version). Nonetheless, to avoid any misinterpretation, in the revised manuscript we further clarified this section. Additionally, following the reviewer's suggestion we have down toned and rewritten the mentioned sentence as follows: *“Thus, these results suggest that Bmal1 depletion leads to transcriptional changes in genes related to developmental processes by direct or indirect mechanisms.”*

Additional Figure 9. Gene set enrichment analysis (GSEA) of the RNA-seq data from wild type (WT) and knock out (KO) *Bmal1* ESCs. Sets of the ESC-enriched genes (Ben-Porath et al., 2008) and the targets of the pluripotency markers POU5F1, SOX2 and NANOG (Ang et al., 2011; Lee et al., 2012) are used.

(2.2) Figure 3: the results in this figure are not clear. Experiments should be explained in more detail and results flushed out.

Following the reviewer's suggestion, we have now improved the section regarding **Figure 3** (in the previous version) by providing more detailed information about the *in silico* analysis using our previously published genome-wide RNAi screening, which was designed to identify factors important in early stages of differentiation [6].

(2.3) Figure 7: BMAL1 KO results in modifications of metabolic gene expression (7A). The authors should clarify the identity of these genes (glycolytic? mitochondrial/TCA etc?). This is important in order to support their next finding. In addition, how do they reconcile reduced glycolysis in BMAL KO ESCs with their differentiation status?

This is another interesting remark from the reviewer. Following his/her suggestion, we provide in the current manuscript version a more detailed analysis of the expression levels of metabolic-related genes affected by *Bmal1* depletion in ESCs (**Additional Figure 10A**) (**Figure 7A**).

On the other hand, although how cellular metabolism at the molecular level directly influences cell fate decisions is an emerging area of study, accumulating evidence have shown active roles of metabolism influencing gene expression that is key for self-renewal and differentiation of pluripotent cells, including mouse and human ESCs [7-11]. In particular, it is well documented that increased OXPHOS and ROS levels accompany the differentiation process [8,10,11]. Indeed, the balance between glycolysis and OXPHOS is critical for modulating the differentiation potential of pluripotent cells [8,10,11]. Importantly, as the reviewer mentioned, our results show that *Bmal1* KO ESCs display higher OXPHOS activity and elevated ROS levels together with altered differentiation potential (i.e. assessed by EB and gastruloid differentiation approaches). Thus, our data are in line with the current literature showing that altering cellular metabolic states in pluripotent cells will affect their differentiation status and link BMAL1 to the metabolic control of differentiation. Nonetheless, to strengthen our findings, we have now provided new data showing that reducing OXPHOS activity by inhibition of mitochondrial respiration (i.e. with Antimycin A/Rotenone), was sufficient to partially rescue the expression of lineage markers which were aberrant induced during mesendoderm differentiation *in vitro*

in *Bmal1* KO ESCs compared to *Bmal1* WT ESCs (**Additional Figure 10B**) (**Figure 7H**).

Additional Figure 10. (A) Heatmap of the expression of the indicated metabolism-related genes in wild type (*Bmal1* WT) and *Bmal1* knock-out (*Bmal1* KO) ESCs. **(B)** Relative expression of the indicated genes in *Bmal1* WT and KO cells with or without Antinomycin A/Rotenone (AA/Rot) treatment. Student's t-test was performed. A representative experiment is shown (n=3 technical replicates). * $P < 0.05$, ** $P < 0.01$, *** $P < 0.001$, **** $P < 0.0001$

(2.4) Figure 4E: BMAL1 KO shows upregulation of ectoderm genes associated with downregulation of endoderm, mesoderm genes. Does loss of BMAL1 enhance the ESC differentiation to ectoderm (at the expense of other germ layers)?

This is an interesting remark from the reviewer. As we mentioned in the text of the previous version of the manuscript: “Indeed, when we analysed gene expression of several ectoderm, mesoderm and endoderm markers, we found that they were differentially induced in *Bmal1* KO at day 6, compared to WT EBs (Fig 4E). These results indicate that *Bmal1* is required for ESC differentiation to properly establish germ layer specific transcriptional programs.” As pointed by the reviewer, *Bmal1* depletion induced ectoderm marker expression and reduced endoderm marker expression both in EB and gastruloid differentiation (Figures 4E and 5E in the previous version). Nevertheless, we cannot definitely conclude that *Bmal1* loss-of-function skews differentiation towards ectoderm at the expense of mesoderm and endoderm germ

layers, given that we observed a differentiation assay-dependent effect of BMAL1 on several ectoderm (i.e. *Sox1*) and mesoderm (i.e. *Cxcl12*, *Mixl1* and *T*) markers. Altogether, from our data we can only conclude that BMAL1 loss influences the differentiation potential of ESCs *in vitro*, and future studies will be required to further characterize in detail the contribution of BMAL1 to the transcriptional regulation taking place during germ layers specification.

Minor:

(2.5) The authors should quantify the bands in WB (Figure 1B, 2B etc).

Following the reviewer's suggestion, we have now quantified all WBs in the previous manuscript version: **Figure 1B (Additional Figure 11) (Figure 1B)** and **Figure 2B (Additional Figure 2) (Figure 2B)**.

Additional Figure 10. (Left) Western Blot of BMAL1 and POU5F1 in ESCs transfected with shRNAs against *Bmal1* or *Luciferase* as control. ACTIN was used as a loading control. (Right) Quantification of the blot.

(2.6) Figure 2D, it is important to show protein expression.

While the protein expression of both NANOG and POU5F1 was already shown in **Figure 2B** (previous version), following the reviewer's suggestion we have now included the western blot analysis of ZFP42 protein abundance in wild type and *Bmal1* KO ESCs (**Additional Figure 2**) (**Figure 2B**).

(2.7) Figure 5C-D: please provide quantitative data measuring the size of the gastruloid structure and possibly the elongation/polarization.

This is another interesting point from the reviewer. This data has been now included in the revised version of the manuscript (please, see above our **response to Reviewer#1.6** and **Additional Figure 6**).

(2.8) How do gene expressions in Figure 5B relate to that in 5E?

In **Figure 5B** (previous version) we depicted the expression level of *Bmal1*, the pluripotency marker *Nanog* and the differentiation markers Pax3, Gata6 and Mixl1 after 120h of gastruloid formation using wild type ESCs (Relative expression is shown as relative to 0h). On the other hand, in **Figure 5E** (previous version) we represent the expression of the indicated genes of interest comparing wild type and *Bmal1* KO cell lines during gastruloid formation (at 0h and 120h). To clarify this aspect, we have now added more detailed information in the figure legend.

(2.9) The authors should explain in detail of how the results from Figure 7F-7G have been generated.

Following the reviewer's suggestion, we have now included more detailed information regarding how proton leak and coupling efficiency (%) were calculated and analysed in the MATERIALS AND METHODS section.

REFERENCES

1. Beccari L, Moris N, Girgin M, Turner DA, Baillie-Johnson P, Cossy AC, Lutolf MP, Duboule D, Arias AM (2018) Multi-axial self-organization properties of mouse embryonic stem cells into gastruloids. *Nature* **562**: 272-276
2. Takahashi JS (2017) Transcriptional architecture of the mammalian circadian clock. *Nat Rev Genet* **18**: 164-179
3. Ben-Porath I, Thomson MW, Carey VJ, Ge R, Bell GW, Regev A, Weinberg RA (2008) An embryonic stem cell-like gene expression signature in poorly differentiated aggressive human tumors. *Nat Genet* **40**: 499-507
4. Ang YS, Tsai SY, Lee DF, Monk J, Su J, Ratnakumar K, Ding J, Ge Y, Darr H, Chang B, *et al.* (2011) Wdr5 mediates self-renewal and reprogramming via the embryonic stem cell core transcriptional network. *Cell* **145**: 183-197
5. Lee DF, Su J, Ang YS, Carvajal-Vergara X, Mulero-Navarro S, Pereira CF, Gingold J, Wang HL, Zhao R, Sevilla A, *et al.* (2012) Regulation of embryonic and induced pluripotency by aurora kinase-p53 signaling. *Cell Stem Cell* **11**: 179-194
6. Gingold JA, Fidalgo M, Guallar D, Lau Z, Sun Z, Zhou H, Faiola F, Huang X, Lee DF, Waghray A, *et al.* (2014) A genome-wide RNAi screen identifies opposing functions of Snai1 and Snai2 on the Nanog dependency in reprogramming. *Mol Cell* **56**: 140-152
7. Reid MA, Dai Z, Locasale JW (2017) The impact of cellular metabolism on chromatin dynamics and epigenetics. *Nat Cell Biol* **19**: 1298-1306
8. Dahan P, Lu V, Nguyen RMT, Kennedy SAL, Teitell MA (2019) Metabolism in pluripotency: Both driver and passenger? *J Biol Chem* **294**: 5420-5429
9. Moussaieff A, Rouleau M, Kitsberg D, Cohen M, Levy G, Barasch D, Nemirovski A, Shen-Orr S, Laevsky I, Amit M, *et al.* (2015) Glycolysis-mediated changes in acetyl-CoA and histone acetylation control the early differentiation of embryonic stem cells. *Cell Metab* **21**: 392-402
10. Zhang J, Zhao J, Dahan P, Lu V, Zhang C, Li H, Teitell MA (2018) Metabolism in Pluripotent Stem Cells and Early Mammalian Development. *Cell Metab* **27**: 332-338
11. Wu J, Ocampo A, Belmonte JCI (2016) Cellular Metabolism and Induced Pluripotency. *Cell* **166**: 1371-1385

MS: LSA-2019-00534-T

Miguel Fidalgo
University of Santiago de Compostela
Physiology
Barcelona Avenue s/n-Campus Vida
Santiago de Compostela, A Corunha 15782
Spain

Dear Dr. Fidalgo,

Thank you for your recent correspondence regarding our decision on your manuscript entitled "BMAL1 coordinates energy metabolism and differentiation of pluripotent stem cells". We appreciate the revision outline provided and will be happy to consider your work further here. Please note that we will need strong support on the revised version from experts. Not addressing pluripotency in vivo may be seen as impeding publication. It is difficult to predict such outcome at this pre-revision stage, but I wanted to be very open about this so that you consider your options carefully.

Yours sincerely,

Andrea Leibfried, PhD
Executive Editor
Life Science Alliance
Meyerohofstr. 1
69117 Heidelberg, Germany
t +49 6221 8891 502
e a.leibfried@life-science-alliance.org
www.life-science-alliance.org

We appreciate the constructive and in-depth comments of the two reviewers, which have helped us to design and perform additional experiments, thus further clarifying the presentation of our data and strengthening the conclusions of our manuscript. Here we have provide detailed point-by-point responses to reviewers' comments together with our additional information and new experimental data. For the reviewers' convenience, we have highlighted in **red** those new data figures that have been incorporated in the revised manuscript and are also mentioned in this rebuttal.

Reviewer #1 (Comments to the Authors (Required)):

In this manuscript, the authors demonstrated that BMAL1 was dispensable for ESC self-renewal, but played a role in the exit of pluripotency and cell differentiation commitment. Mechanistically, the authors illustrated that BMAL1 participated in regulating energy metabolism and mitochondrial function. Overall, the part that BMAL1 was involved in the metabolic regulation is interesting.

We thank the reviewer for his/her positive comments and appreciation of our work.

However, the other parts of this paper are not very conclusive due to either lack of statistical analysis or bad quality of the data. More experiments are needed to clarify these issues.

We apologize for having forgotten to perform the statistical analysis of some panels in the first version of the manuscript. We have now fixed this in the revised current version of the manuscript. On the other hand, we respectfully disagree with the reviewer on the comment that "*bad quality of the data*" and we think this may be due to a confusion (please see below point no. #1.2).

Concerns:

(1.1) In Fig1A, the authors examined the RNA level of Bmal1, Clock, Nanog and Pou5f1 in MEFs and ESCs. Since RNA level could not fully represent protein level, western blotting analysis of these proteins and statistical analysis is required.

This is a good suggestion from the reviewer. Indeed, we have followed his/her indications and analysed the protein expression levels of BMAL1, NANOG and POU5F1 (**Additional Figure 1A**) (**Figure 1A**). Importantly, consistent with the RNA expression (**Figure 1A** in original manuscript version) we found that BMAL1 protein levels are also significantly higher in pluripotent ESCs than in somatic MEFs (**Additional Figure 1B**) (**Figure 1A**).

Additional Figure 1. (A) Western blot of BMAL1, NANOG and POU5F1 in mouse embryonic fibroblasts (MEFs) and embryonic stem cells (ESCs). β TUBULIN was used as loading control. Biological triplicates are shown. Quantification of the blots is shown in **(B)**. Bars represent mean \pm s.d. * P <0.05, *** P <0.001, **** P <0.0001 significantly differences calculated using two-tailed unpaired Student's t -test analysis.

(1.2) The quality of western blotting in Fig2A and Fig 2B are bad. Better quality of western blotting and statistical analysis of Nanog or the other markers should be performed.

We respectfully disagree with the reviewer and as stated above, we think that there is a confusion when he/she states that “the quality of western blotting in Fig2A and Fig 2B are bad” because:

- i) **Figure 2A** in the previous version of the manuscript was not a Western Blot (WB), but instead it showed a schematic representation for the CRISPR/Cas9 strategy used in our study.
- ii) The WB showed in **Figure 2B** in the previous version showed unique and clear bands for each of the antibody used (i.e. BMAL1, NANOG, POU5F1, ACTB).

Nevertheless, following the reviewer’s suggestion we have now provided new data that replaces **Figure 2B** from the previous manuscript version, with biological triplicates and statistical analysis (**Additional Figure 2**) (**Figure 2B**).

Additional Figure 2. (A) Western Blot of BMAL1 and pluripotent factors ZFP42, NANOG and POU5F1 in *Bmal1* WT and KO ESCs. ACTIN was used as a loading control. **(B)** Quantification of three independent experiments. Bars represent mean \pm s.d. ** P <0.01 significantly differences calculated using two-tailed unpaired Student's t -test analysis. ns: not significant.

(1.3) In Fig2A, it should be 22 bp deletion in Bmal1 KO ESCs instead of 20 bp. We apologize that in our current version of the manuscript we mislabeled the deletion as remarked by the reviewer. Now we have fixed this issue (**Additional Figure 3**) (**Figure 2A**).

Additional Figure 3. (Top) Schematic representation for the CRISPR/Cas9 strategy used. The designed sgRNA is underlined and the PAM sequence is highlighted in red. (Bottom) Sequence deleted (22 bp) in *Bmal1* KO alleles detected by SANGER sequencing.

(1.4) In the text, the authors claimed that the self-renewal of ESCs was not greatly affected by deletion of BMAL1 neither on colony number nor on the colony morphology (Fig 2F and 2G). However, in Fig2G, it seems that knockout of Bmal1 affects the differentiation. Convincing conclusion should be stated. In addition, statistical analysis is missing in Fig 1F.

We would like to apologize if our choice of words when we said “was not greatly affected” could generate confusion. Following the reviewer’s recommendation and to avoid any misinterpretation, we have now rephrased it indicating the specific percentages in the text as follows: “On the other hand, when we analysed the self-renewal capability of BMAL1-depleted ESCs we did not observe changes in the number of colonies (Figure 2F) and only a mild reduction in the percentage of undifferentiated colonies (74.5±5.6 versus 59.7±5.7) in presence of LIF (Figure 2G), which is in line with the acute silencing of Bmal1 using the shRNA approach (Figure 1F). Of note, Bmal1 KO ESCs showed a defect in differentiation upon LIF withdrawal as observed by the decrease in the percentage of fully differentiated colonies (65.2±4.2 versus 34.5±3.1) (Figure 2G), suggesting a possible role of BMAL1 during the exit of pluripotency.”

On the other hand, to address his/her suggestion regarding to lack of statistical analysis in **Figure 1F** in the previous version, we have now added that piece of information in the current revised manuscript (**Additional Figure 4**) (**Figure 1F**).

Additional Figure 4. ESCs transfected with shRNA against *Bmal1* or *Luciferase* as control are seeded at low confluence in standard medium with or without LIF, and colonies are counted and classified into the three indicated categories according to their AP staining intensity (n=3). Bars represent mean \pm s.e.m. Significant differences calculated using two-tailed unpaired Student's *t*-test analysis. ns: not significant.

(1.5) ESC pluripotency analysis is required in *in vivo* experiment to check whether BMAL1 KO ESCs can form the three germ layers.

We thank the reviewer for his/her suggestion. Following the reviewer's suggestion we have now analysed the ability of *Bmal1* KO ESCs to form the three layers *in vivo* using teratoma assays. For that purpose, eight female Swiss Nude mice at 5 weeks of age were injected subcutaneously in their flanks with wild type or *Bmal1* KO ESCs. Teratomas were excised 4 weeks after injection, measured and processed for staining with hematoxylin and eosin (H&E). Our results show that *Bmal1* KO ESCs can form teratomas containing differentiated tissues from the three germinal layers (Additional Figure 5) (Figures 4A and 4B). Notably, this data is consistent with our results in the previous manuscript version, which show that *Bmal1* KO ESCs are pluripotent given that: i) BMAL1 is dispensable for the maintenance of the pluripotent cellular state (Figures 1 and 2 in the previous version); ii) even though presence of BMAL1 *in vitro* is required for proper expression of the lineage markers of ectoderm, mesoderm and endoderm germ layers (Figure 4E in the previous version), *Bmal1* KO ESCs are capable of forming embryoid bodies (EBs) (Figures 4C and 4D in the previous version). Altogether, our data support that *Bmal1* KO ESCs are pluripotent and that loss of *Bmal1* leads to defects that interfere with normal expression patterns of lineage specification markers, which are detectable during *in vitro* differentiation. However, we cannot discard the possibility that during *in vivo* differentiation (i.e. by teratoma assays) there are also aberrant expression of specific markers of one or more germ layers even though *Bmal1* KO teratomas seem grossly normal.

Additional Figure 5. (A) Teratomas formed by wild type (WT) and *Bmal1* knock-out (KO) ES cells after 3-4 weeks of injection. (B) Quantification of teratomas size (8 tumors per genotype). Data are shown as mean±s.d. Student's *t*-test was performed. ns: not significant. (C) Histological analysis of teratomas of the indicated genotype by H&E staining. Structures representing the three embryonic germ layers in both genotypes were found. (*): connective tissue. (^): neural tissue and (+): epithelial tissue. Image magnification: 20X.

(1.6) In Fig 5D, BMAL1 deficiency resulted in abnormal *in vitro* gastrulation, which lacks statistical analysis. More evidences are needed to confirm the relationship between aberrant upregulation of three-germ-layer markers and the deficiency of polarization or elongation during *in vitro* gastrulation.

We apologize for not having included the statistical analysis in **Figure 5D** in the previous version of the manuscript. Following the reviewer's suggestion we have now performed additional gastruloid formation assays and included statistical analysis (**Additional Figure 6A**) (**Figure 5E**). Consistent with our previous data, we found that absence of BMAL1 significantly impact in gastruloid formation efficiency compared to wild-type (7.14% vs 41.07%, respectively). Additionally, to further characterize the effect of *Bmal1* depletion in elongation during *in vitro* gastrulation, we have now analysed the size of *Bmal1* WT or KO ESC-derived aggregates during gastruloid formation assay (**Additional Figure 6B**) (**Figure 5D**). Our data supports that absence of BMAL1 impairs the elongation process *in vitro*, giving rise to cellular aggregates displaying reduced size at 120 hours. Likewise, we observed altered transcriptional dynamics of several members of the *Hoxd* gene cluster (**Additional Figure 6C**) (**Figure 5G**), which is one of the hallmarks of axial gene regulatory systems whose sequential activation is associated with the patterning and formation during *in vitro* gastruloid formation by the seminal work of Beccari and collaborators [1]. Thus, these results are in line with the observed deregulation, in absence of BMAL1, of some germ layer markers (**Figures 5E** in the previous version) that have been reported to be timely transcribed during *in vitro* gastruloid formation [1] (i.e. *Sox1* and *Pax3* which should be localized centrally; *Gata4*, *Gata6* and

T/Brachyury expressed in post-occipital structures; *Mixl1* and *Eomes* associated with the gastrulation process *in vivo*).

Additional Figure 6. (A) Percentage of gastruloid-like structures formed by *Bmal1* WT and KO aggregates at 120h of *in vitro* gastrulation. (n=14 independent experiments with at least 8 formed aggregates per experiment). Bars represent mean \pm s.e.m. Two-tailed unpaired Student's *t*-test was calculated. **** $P < 0.0001$. **(B)** Quantification of the size of the aggregates formed by *Bmal1* WT and KO cells at 120h of *in vitro* gastrulation. a.u.: arbitrary units. All data are shown as mean \pm s.e.m. Two-tailed unpaired Student's *t*-test was calculated. **** $P < 0.0001$. **(C)** Expression of *Hoxd1*, *Hoxd3*, *Hoxd4* and *Hoxd10* dynamics during *in vitro* gastrulation of *Bmal1* WT and KO ESCs at the indicated timepoints. (n=3). Data is represented as mean \pm s.e.m.

(1.7) The authors mentioned that close correlation of the transcriptional changes of regulators related with cellular differentiation and cellular metabolism including *Zfx* and Polycomb members (i.e. *Eed* and *Suz12*) were found upon depletion of *Bmal1* and *Pou5f1* (Fig EV3A). Experiments including RT-qPCR or western blotting of these regulators in *Bmal1* KO and *Pou5f1* KO cells should be performed to support the conclusion.

We apologize if in the previous version of the manuscript the message in the **“Results section: *Bmal1* Supports Glycolytic Metabolism in ESCs”** that we tried to convey was not clear enough and led the reviewer to a misunderstanding. What we meant by: “we found a close correlation of the transcriptional changes upon *Bmal1* depletion with those upon depletion of the master regulator of pluripotency *Pou5f1* [25-27] as well as other pluripotent regulators related with cellular differentiation and cellular metabolism including *Zfx* [28,29] and Polycomb members (i.e. *Eed* and *Suz12*) [30,31] (Fig EV3A)” was that depletion of *Bmal1*, produces similar transcriptional changes (or at least in part correlate with) to the ones previously reported after depletion of *Pou5f1* or *Zfx* or *Eed* or *Suz12* in mouse ESCs.

Nevertheless, following the reviewer’s suggestion we have now assessed the effect of *Bmal1* depletion in the abundance of *Pou5f1*, *Zfx*, *Eed* or *Suz12* (**Additional Figure 7**) (**Figures EV3C and EV3D**). Our results show that *Bmal1* depletion does not alter the expression of *Pou5f1*, *Zfx*, *Eed* or *Suz12* in mouse ESCs.

In order to avoid any possible misperception or confusion from readers we have now clarified this in the text as follows: “To understand how *BMAL1* orchestrates transcriptional programs involved in proper cell differentiation in vitro, we next compared the gene expression changes after *Bmal1* depletion in ESCs with those caused by loss-of-function of other transcription regulators using the Network2Canvas computational tool (see Materials and Methods). Interestingly, we found a close correlation of the transcriptional changes in absence of *BMAL1* with those upon depletion of the master pluripotency regulator *Pou5f1* [25-27] as well as other pluripotent regulators related with differentiation and/or cellular metabolism including *ZFX* [28,29] and Polycomb members (i.e. *EED* and *SUZ12*) [30,31] (**Figure EV3B**). These results, together with the observation that the expression of these factors was not altered upon *Bmal1* depletion in mouse ESCs (**Figures EV3C and EV3D**), suggest the existence of common pathways regulated by this set of factors and *BMAL1*.”

Additional Figure 7. (A) RNA expression of the indicated genes in *Bmal1* wild type (WT) and knock-out (KO) ESCs detected by RNA-seq. FPKM: fragments per million mapped reads. **(B)** Western Blot of Polycomb proteins (EZH2 and SUZ12) and the pluripotency master regulator POU5F1 in *Bmal1* WT and KO ESCs. GAPDH was used as a loading control.

(1.8) Scale bar is missing in most of the staining data, such as in Fig1C, 1D, 5C, 7H, EV1B. We apologize as the scale bar size that we used in the previous manuscript version was too small and lead the confusion that they were not present. In order to improve the clarity of the representation, we have now increased the size of the scale bars in all figures across the current manuscript version (**Additional Figure 8**).

Additional Figure 8. Figures numbered as in the previous version of the manuscript are shown. (**Figure 1C**) Bright Field (BF) and Alkaline-phosphatase staining (AP) images of ESCs transfected with shRNA against *Bmal1* or Luciferase as a control. White bars represent 200 μ m. (**Figure 1D**) Immunofluorescence of the pluripotent marker SSEA1 in ESCs transfected with shRNAs against *Bmal1* or Luciferase as control. Nuclei were stained with DAPI. White bars represent 50 μ m. (**Figure 4C**) Representative images of embryoid bodies at day 6 of differentiation in the indicated cell lines. White bars represent 200 μ m. (**Figure 5C**) Images of 120h aggregates for *Bmal1* WT and KO cells. White bars represent 200 μ m. (**Figure 7H**) Fluorescence representative microscopy images showing MitoSOX fluorescence (red) in WT and *Bmal1* KO ESCs. Nuclei were stained with DAPI and shown in blue. White bars represent 50 μ m. (**Figure EV1B**) Immunofluorescence of pluripotent markers POU5F1, and ZFP281 in ESCs transfected with shRNAs against *Bmal1* or Luciferase as control. Nuclei were stained with DAPI. White bars represent 100 μ m.

Reviewer #2 (Comments to the Authors (Required)):

The studies by Ameneiro and colleagues investigate the function of circadian gene BMAL1 in regulating embryonic stem cells (ESC) differentiation and metabolism. They first show that BMAL1 is dispensable for ESC pluripotency. They further show using both RNA interference and CRISPR-Cas9 that loss of BMAL1 does not significantly alter pluripotency in ESCs. Then through a set of experiments they propose that BMAL1 may be involved in early decisions during pluripotency exit. They further provide evidence that loss of BMAL1 impairs ESC differentiation and alters ESC gene expression (upregulation of ectoderm vs downregulation of mesoderm and endoderm genes). They also use in vitro gastruloid formation to show that BMAL1 KO cells do not elongate properly in vitro. Finally, they show that loss of BMAL1 is associated with metabolic alterations. This manuscript address an important question regarding BMAL1 regulation of ESC fate determination. The authors have performed enormous amount of work using a combination of approaches and have produced interesting data. However, their conclusion in some instances is premature or not supported by the data.

We thank the reviewer for these positive comments on our manuscript.

My comments are as follow:

(2.1) Figure 6: please clarify how this figure supports BMAL1 control of transcriptional regulation of ESC differentiation. As the authors have noted loss of BMAL1 results in modulations of cellular status (pluripotency vs differentiation). The transcriptomic changes may be just a reflection of cellular differentiation and not the cause. The data does not support: "Thus, these results show that BMAL1 is involved in transcriptional regulation of developmental processes in ESCs" as stated by the authors.

We appreciate the reviewer's comment and we apologize for the overstatement that "*these results show that BMAL1 is involved in transcriptional regulation of developmental processes in ESCs*". It is true that while BMAL1 has been widely reported to function as a transcriptional regulator in other cellular contexts (Reviewed in [2]), in this manuscript we have not directly addressed whether BMAL1 regulates in a direct or indirect manner the transcription of the 444 upregulated and 197 downregulated genes (**Figure 6A** in the previous version). This is an interesting question to be addressed by us or other laboratories in future studies. On the other hand, depletion of *Bmal1* (i.e. KD or KO) in ESCs does not affect:

- a) their self-renewal ability in presence of LIF (**Figures 1G-F and 2F-G** in the previous version)
- b) the expression of key pluripotency gene regulators such as *Pouf51* (also known as *Oct4*), *Nanog*, *Sox2* are the RNA (**Figures EV1A and 2D**) or protein levels (**Figures 1B, 1D, EV1B, 2B, 2E and EV2D** in the previous manuscript version)

Taking into account these data, the transcriptomic changes observed upon *Bmal1* depletion in ESCs in presence of LIF (**Figure 6** in the previous version) cannot be just a reflection of the

cellular differentiation status of the cells. Additionally, we have now evaluated whether *Bmal1* depletion in ESCs affects the expression of pluripotency enriched genes [3] as well as the targets of the master regulators of pluripotency POU5F1 (OCT4), SOX2 and NANOG (OSN) [4,5] by gene set enrichment analysis (GSEA) (**Additional Figure 9**) (**Figure EV3A**). Similar to our results mentioned above, absence of BMAL1 does not significantly impact the expression of ESC-enriched genes or OSN targets, further indicating that *Bmal1* KO ESCs remain overall pluripotent. Thus, our results suggest that the transcriptional changes in differentiation-related markers in ESCs in absence of BMAL1 are not sufficient to trigger differentiation under pluripotent cell culture conditions (i.e. presence of LIF). However, transcriptional deregulation of lineage-specific markers in *Bmal1* KO ESCs (**Figures 6B-F** in the previous version) are in line with our data showing that depletion of *Bmal1* greatly affects the *in vitro* differentiation potential of ESCs (**Figures 4 and 5** in the previous version). Nonetheless, to avoid any misinterpretation, in the revised manuscript we further clarified this section. Additionally, following the reviewer's suggestion we have down toned and rewritten the mentioned sentence as follows: *“Thus, these results suggest that Bmal1 depletion leads to transcriptional changes in genes related to developmental processes by direct or indirect mechanisms.”*

Additional Figure 9. Gene set enrichment analysis (GSEA) of the RNA-seq data from wild type (WT) and knock out (KO) *Bmal1* ESCs. Sets of the ESC-enriched genes (Ben-Porath et al., 2008) and the targets of the pluripotency markers POU5F1, SOX2 and NANOG (Ang et al., 2011; Lee et al., 2012) are used.

(2.2) Figure 3: the results in this figure are not clear. Experiments should be explained in more detail and results flushed out.

Following the reviewer's suggestion, we have now improved the section regarding **Figure 3** (in the previous version) by providing more detailed information about the *in silico* analysis using our previously published genome-wide RNAi screening, which was designed to identify factors important in early stages of differentiation [6].

(2.3) Figure 7: BMAL1 KO results in modifications of metabolic gene expression (7A). The authors should clarify the identity of these genes (glycolytic? mitochondrial/TCA etc?). This is important in order to support their next finding. In addition, how do they reconcile reduced glycolysis in BMAL KO ESCs with their differentiation status?

This is another interesting remark from the reviewer. Following his/her suggestion, we provide in the current manuscript version a more detailed analysis of the expression levels of metabolic-related genes affected by *Bmal1* depletion in ESCs (**Additional Figure 10A**) (**Figure 7A**).

On the other hand, although how cellular metabolism at the molecular level directly influences cell fate decisions is an emerging area of study, accumulating evidence have shown active roles of metabolism influencing gene expression that is key for self-renewal and differentiation of pluripotent cells, including mouse and human ESCs [7-11]. In particular, it is well documented that increased OXPHOS and ROS levels accompany the differentiation process [8,10,11]. Indeed, the balance between glycolysis and OXPHOS is critical for modulating the differentiation potential of pluripotent cells [8,10,11]. Importantly, as the reviewer mentioned, our results show that *Bmal1* KO ESCs display higher OXPHOS activity and elevated ROS levels together with altered differentiation potential (i.e. assessed by EB and gastruloid differentiation approaches). Thus, our data are in line with the current literature showing that altering cellular metabolic states in pluripotent cells will affect their differentiation status and link BMAL1 to the metabolic control of differentiation. Nonetheless, to strengthen our findings, we have now provided new data showing that reducing OXPHOS activity by inhibition of mitochondrial respiration (i.e. with Antimycin A/Rotenone), was sufficient to partially rescue the expression of lineage markers which were aberrant induced during mesendoderm differentiation *in vitro*

in *Bmal1* KO ESCs compared to *Bmal1* WT ESCs (**Additional Figure 10B**) (**Figure 7H**).

Additional Figure 10. (A) Heatmap of the expression of the indicated metabolism-related genes in wild type (*Bmal1* WT) and *Bmal1* knock-out (*Bmal1* KO) ESCs. **(B)** Relative expression of the indicated genes in *Bmal1* WT and KO cells with or without Antinomycin A/Rotenone (AA/Rot) treatment. Student's t-test was performed. A representative experiment is shown (n=3 technical replicates). * $P < 0.05$, ** $P < 0.01$, *** $P < 0.001$, **** $P < 0.0001$

(2.4) Figure 4E: BMAL1 KO shows upregulation of ectoderm genes associated with downregulation of endoderm, mesoderm genes. Does loss of BMAL1 enhance the ESC differentiation to ectoderm (at the expense of other germ layers)?

This is an interesting remark from the reviewer. As we mentioned in the text of the previous version of the manuscript: "Indeed, when we analysed gene expression of several ectoderm, mesoderm and endoderm markers, we found that they were differentially induced in *Bmal1* KO at day 6, compared to WT EBs (Fig 4E). These results indicate that *Bmal1* is required for ESC differentiation to properly establish germ layer specific transcriptional programs." As pointed by the reviewer, *Bmal1* depletion induced ectoderm marker expression and reduced endoderm marker expression both in EB and gastruloid differentiation (Figures 4E and 5E in the previous version). Nevertheless, we cannot definitely conclude that *Bmal1* loss-of-function skews differentiation towards ectoderm at the expense of mesoderm and endoderm germ

layers, given that we observed a differentiation assay-dependent effect of BMAL1 on several ectoderm (i.e. *Sox1*) and mesoderm (i.e. *Cxcl12*, *Mixl1* and *T*) markers. Altogether, from our data we can only conclude that BMAL1 loss influences the differentiation potential of ESCs *in vitro*, and future studies will be required to further characterize in detail the contribution of BMAL1 to the transcriptional regulation taking place during germ layers specification.

Minor:

(2.5) The authors should quantify the bands in WB (Figure 1B, 2B etc).

Following the reviewer's suggestion, we have now quantified all WBs in the previous manuscript version: **Figure 1B (Additional Figure 11) (Figure 1B)** and **Figure 2B (Additional Figure 2) (Figure 2B)**.

Additional Figure 10. (Left) Western Blot of BMAL1 and POU5F1 in ESCs transfected with shRNAs against *Bmal1* or *Luciferase* as control. ACTIN was used as a loading control. (Right) Quantification of the blot.

(2.6) Figure 2D, it is important to show protein expression.

While the protein expression of both NANOG and POU5F1 was already shown in **Figure 2B** (previous version), following the reviewer's suggestion we have now included the western blot analysis of ZFP42 protein abundance in wild type and *Bmal1* KO ESCs (**Additional Figure 2) (Figure 2B)**.

(2.7) Figure 5C-D: please provide quantitative data measuring the size of the gastruloid structure and possibly the elongation/polarization.

This is another interesting point from the reviewer. This data has been now included in the revised version of the manuscript (please, see above our **response to Reviewer#1.6** and **Additional Figure 6)**.

(2.8) How do gene expressions in Figure 5B relate to that in 5E?

In **Figure 5B** (previous version) we depicted the expression level of *Bmal1*, the pluripotency marker *Nanog* and the differentiation markers Pax3, Gata6 and Mixl1 after 120h of gastruloid formation using wild type ESCs (Relative expression is shown as relative to 0h). On the other hand, in **Figure 5E** (previous version) we represent the expression of the indicated genes of interest comparing wild type and *Bmal1* KO cell lines during gastruloid formation (at 0h and 120h). To clarify this aspect, we have now added more detailed information in the figure legend.

(2.9) The authors should explain in detail of how the results from Figure 7F-7G have been generated.

Following the reviewer's suggestion, we have now included more detailed information regarding how proton leak and coupling efficiency (%) were calculated and analysed in the MATERIALS AND METHODS section.

REFERENCES

1. Beccari L, Moris N, Girgin M, Turner DA, Baillie-Johnson P, Cossy AC, Lutolf MP, Duboule D, Arias AM (2018) Multi-axial self-organization properties of mouse embryonic stem cells into gastruloids. *Nature* **562**: 272-276
2. Takahashi JS (2017) Transcriptional architecture of the mammalian circadian clock. *Nat Rev Genet* **18**: 164-179
3. Ben-Porath I, Thomson MW, Carey VJ, Ge R, Bell GW, Regev A, Weinberg RA (2008) An embryonic stem cell-like gene expression signature in poorly differentiated aggressive human tumors. *Nat Genet* **40**: 499-507
4. Ang YS, Tsai SY, Lee DF, Monk J, Su J, Ratnakumar K, Ding J, Ge Y, Darr H, Chang B, *et al.* (2011) Wdr5 mediates self-renewal and reprogramming via the embryonic stem cell core transcriptional network. *Cell* **145**: 183-197
5. Lee DF, Su J, Ang YS, Carvajal-Vergara X, Mulero-Navarro S, Pereira CF, Gingold J, Wang HL, Zhao R, Sevilla A, *et al.* (2012) Regulation of embryonic and induced pluripotency by aurora kinase-p53 signaling. *Cell Stem Cell* **11**: 179-194
6. Gingold JA, Fidalgo M, Guallar D, Lau Z, Sun Z, Zhou H, Faiola F, Huang X, Lee DF, Waghray A, *et al.* (2014) A genome-wide RNAi screen identifies opposing functions of Snai1 and Snai2 on the Nanog dependency in reprogramming. *Mol Cell* **56**: 140-152
7. Reid MA, Dai Z, Locasale JW (2017) The impact of cellular metabolism on chromatin dynamics and epigenetics. *Nat Cell Biol* **19**: 1298-1306
8. Dahan P, Lu V, Nguyen RMT, Kennedy SAL, Teitell MA (2019) Metabolism in pluripotency: Both driver and passenger? *J Biol Chem* **294**: 5420-5429
9. Moussaieff A, Rouleau M, Kitsberg D, Cohen M, Levy G, Barasch D, Nemirovski A, Shen-Orr S, Laevsky I, Amit M, *et al.* (2015) Glycolysis-mediated changes in acetyl-CoA and histone acetylation control the early differentiation of embryonic stem cells. *Cell Metab* **21**: 392-402
10. Zhang J, Zhao J, Dahan P, Lu V, Zhang C, Li H, Teitell MA (2018) Metabolism in Pluripotent Stem Cells and Early Mammalian Development. *Cell Metab* **27**: 332-338
11. Wu J, Ocampo A, Belmonte JCI (2016) Cellular Metabolism and Induced Pluripotency. *Cell* **166**: 1371-1385

March 23, 2020

Re: Life Science Alliance manuscript #LSA-2019-00534-TR-A

Dr. Miguel Fidalgo
University of Santiago de Compostela
Physiology
Barcelona Avenue s/n-Campus Vida
Santiago de Compostela, A Corunha 15782
Spain

Dear Dr. Fidalgo,

Thank you for submitting your manuscript entitled "BMAL1 coordinates energy metabolism and differentiation of pluripotent stem cells" to Life Science Alliance. One of the original reviewers evaluated your revised manuscript. As you will see, the reviewer still raises significant concerns that preclude publication here at this stage.

As you know, we only allow one round of experimental revisions. However, should you be prepared to address all remaining concerns, we can exceptionally offer to further consider your manuscript for publication. Please let us know how you would like to proceed.

We are aware that many laboratories cannot function fully during the current COVID-19/SARS-CoV-2 pandemic and therefore encourage you to take the time necessary to revise the manuscript to the extent requested above.

Should you decide to further revise your work, please also pay attention to the following:

- Please make sure that the author order in our submission system matches the one depicted in the manuscript text
- Please provide a „summary blurb" within our submission system
- Please upload all figure files, including supplementary figures, as separate ones
- Please add a callout to figure 5C in the manuscript text
- Please note that we only have supplementary figures and tables (currently named expanded view in your manuscript); please adapt
- Figure EV1 (should be Figure S1) should have a callout for panel C (spelling mistake)
- Please provide the manuscript text in docx format
- Please add a scale bare to Figure 4B

Thank you for this interesting contribution to Life Science Alliance. We are looking forward to receiving your revised manuscript.

Sincerely,

B. MANUSCRIPT ORGANIZATION AND FORMATTING:

We encourage our authors to provide original source data, particularly uncropped/-processed electrophoretic blots and spreadsheets for the main figures of the manuscript. If you would like to add source data, we would welcome one PDF/Excel-file per figure for this information. These files

will be linked online as supplementary "Source Data" files.

Reviewer #1 (Comments to the Authors (Required)):

This resubmitted version has addressed some of major concerns from reviewers and the authors have performed a number of additional assays according to suggestions or criticisms by the reviewers, which provides better support for the authors' claims that BMAL1 coordinates energy metabolism and differentiation of pluripotent stem cells. However, there are still some concerns which are related to improvement of the quality and presentation of the data to make the conclusions more convincing.

Major concerns:

1. Although the author argued about the quality of western blotting, the bands of ZFP42 in Fig 2B ran into a line, which makes it difficult to quantify the intensity. The blotting of actin also seems uneven leading to inaccurate statistics. In addition, the blotting in Fig 1B looks like over-adjusted contrast. And at least three times blotting should be performed to obtain statistics. As main figures they seem a little sloppy and better quality of blotting should be provided.
2. To confirm the regulatory effect of BMAL1 on Nanog promoter activity, the quantifications of in Fig 3C should be provided. To further confirm that BMAL1 directly binds to Nanog promoter, ChIP-qPCR should also be performed.
3. The two photographs of BMAL1 WT and BMAL1 KO teratoma in Fig 4A differ significantly in magnification.
4. The structure of the three germ layers in Fig 4B is unclear. Three germ layers should be labelled with specific markers of mesoderm, ectoderm and endoderm.
5. Statistical analysis should be performed in Fig 5G.
6. The RNA-seq heatmap results in Figure 7A show that the repeatability of some genes in the two WT and BMAL1 KO groups is not high. Based on this, it is not reliable to conclude that BMAL1 regulates the mitochondrial complex I-V and the tricarboxylic acid cycle (TCA or Krebs cycle). It is better to present with selected high repeatability genes in mitochondrial complex I-V and TCA and confirm some of them using RT-qPCR.

We would like to thank the reviewer for his/her positive comments on our work, and his/her feedback, which has helped further clarify and enhance our manuscript. Here we provide our responses to each specific comment and suggestion raised by the reviewer and the editor.

Reviewer #1 (Comments to the Authors (Required)):

This resubmitted version has addressed some of major concerns from reviewers and the authors have performed a number of additional assays according to suggestions or criticisms by the reviewers, which provides better support for the authors' claims that BMAL1 coordinates energy metabolism and differentiation of pluripotent stem cells. However, there are still some concerns which are related to improvement of the quality and presentation of the data to make the conclusions more convincing.

Major concerns:

1. Although the author argued about the quality of western blotting, the bands of ZFP42 in Fig 2B ran into a line, which makes it difficult to quantify the intensity. The blotting of actin also seems uneven leading to inaccurate statistics. In addition, the blotting in Fig 1B looks like over-adjusted contrast. And at least three times blotting should be performed to obtain statistics. As main figures they seem a little sloppy and better quality of blotting should be provided.

Following the reviewer's suggestion we provide now a better ZFP42 blot in **Fig S2E** and have replaced ACTIN by β TUBULIN as a loading control in **Fig 2B**. On the other hand, we realized that during the conversion of the figures to .pdf format there was a decrease in the quality of some panels which could give a bad impression. We now provide better image quality for all figures including the Western Blots, and we also provide statistics from three independent experiments for **Fig 1B**.

2. To confirm the regulatory effect of BMAL1 on Nanog promoter activity, the quantifications of in Fig 3C should be provided. To further confirm that BMAL1 directly binds to Nanog promoter, ChIP-qPCR should also be performed.

We provide now the quantification of three independent experiments in **Fig 3D**, further supporting that BMAL1 loss significantly decreases the percentage of *Nanog*-GFP positive cells under mild retinoic acid-induced differentiation conditions. Thus, these results identify BMAL1 as a regulator of *Nanog* expression during the exit of pluripotency. However, as we mention in the current manuscript version, future studies are warranted to investigate whether BMAL1 regulates *Nanog* promoter activity in a direct or indirect manner.

3. The two photographs of BMAL1 WT and BMAL1 KO teratoma in Fig 4A differ significantly in magnification.

We apologize for this, and we have now fixed this issue.

4. The structure of the three germ layers in Fig 4B is unclear. Three germ layers should be labelled with specific markers of mesoderm, ectoderm and endoderm.

We apologize for the bad quality of the final **Fig 4B** after pdf conversion. Now, in the revised manuscript we have improved it, to better appreciate that *Bmal1* KO ESCs can contribute to representative structures of the three germ layers.

5. Statistical analysis should be performed in Fig 5G.

We provide now statistical analysis in **Fig 5G**.

6. The RNA-seq heatmap results in Figure 7A show that the repeatability of some genes in the two WT and BMAL1 KO groups is not high. Based on this, it is not reliable to conclude that BMAL1 regulates the mitochondrial complex I-V and the tricarboxylic acid cycle (TCA or Krebs cycle). It is better to present with selected high repeatability genes in mitochondrial complex I-V and TCA and confirm some of them using RT-qPCR.

In the current manuscript version we provide a new heatmap in **Fig 7A** and validation of some of them in **Fig S4C** following the reviewer's suggestion.

Editor:

- Please make sure that the author order in our submission system matches the one depicted in the manuscript text

We have made sure that the author order both in the submission system and the manuscript is the same.

- Please provide a „summary blurb" within our submission system

We now provide a summary blurb in the submission system.

- Please upload all figure files, including supplementary figures, as separate ones

Main and supplementary figures have been uploaded as individual TIFF files.

- Please add a callout to figure 5C in the manuscript text

We added a callout to Fig 5C in the text in page 8.

- Please note that we only have supplementary figures and tables (currently named expanded view in your manuscript); please adapt

We have fixed this accordingly.

- Figure EV1 (should be Figure S1) should have a callout for panel C (spelling mistake)

This has been corrected in page 32.

- Please provide the manuscript text in docx format

The manuscript text has been provided in docx format.

- Please add a scale bare to Figure 4B

The scale bars in Fig 4B have now been added.

March 31, 2020

RE: Life Science Alliance Manuscript #LSA-2019-00534-TRR

Dr. Miguel Fidalgo
University of Santiago de Compostela
Physiology
Barcelona Avenue s/n-Campus Vida
Santiago de Compostela, A Corunha 15782
Spain

Dear Dr. Fidalgo,

Thank you for submitting your revised manuscript entitled "BMAL1 coordinates energy metabolism and differentiation of pluripotent stem cells". We appreciate the introduced changes and would thus be happy to publish your paper in Life Science Alliance. Before sending you the official acceptance letter, please log in one more time to move all files to the next manuscript version and to fill in the electronic license to publish associated with that version.

A. FINAL FILES:

B. MANUSCRIPT ORGANIZATION AND FORMATTING:

Sincerely,

April 1, 2020

RE: Life Science Alliance Manuscript #LSA-2019-00534-TRRR

Dr. Miguel Fidalgo
University of Santiago de Compostela
Physiology
Barcelona Avenue s/n-Campus Vida
Santiago de Compostela, A Corunha 15782
Spain

Dear Dr. Fidalgo,

Thank you for submitting your Research Article entitled "BMAL1 coordinates energy metabolism and differentiation of pluripotent stem cells". It is a pleasure to let you know that your manuscript is now accepted for publication in Life Science Alliance. Congratulations on this interesting work.

DISTRIBUTION OF MATERIALS:

Again, congratulations on a very nice paper. I hope you found the review process to be constructive and are pleased with how the manuscript was handled editorially. We look forward to future exciting submissions from your lab.

Sincerely,
